# MolSpectra: Pre-training 3D Molecular Representation with Multi-modal Energy Spectra

**Liang Wang**[1,2*] **Shaozhen Liu**[1] **Yu Rong**[3†] **Deli Zhao**[3] **Qiang Liu**[1,2†] **Shu Wu**[1,2] **Liang Wang**[1,2]

[1]New Laboratory of Pattern Recognition (NLPR),
State Key Laboratory of Multimodal Artificial Intelligence Systems (MAIS),
Institute of Automation, Chinese Academy of Sciences (CASIA)
[2]School of Artificial Intelligence, University of Chinese Academy of Sciences
[3]DAMO Academy, Alibaba Group

## Abstract

Establishing the relationship between 3D structures and the energy states of molecular systems has proven to be a promising approach for learning 3D molecular representations. However, existing methods are limited to modeling the molecular energy states from classical mechanics. This limitation results in a significant oversight of quantum mechanical effects, such as *quantized (discrete) energy level structures*, which offer a more accurate estimation of molecular energy and can be experimentally measured through energy spectra. In this paper, we propose to utilize the energy spectra to enhance the pre-training of 3D molecular representations (MolSpectra), thereby infusing the knowledge of quantum mechanics into the molecular representations. Specifically, we propose SpecFormer, a multi-spectrum encoder for encoding molecular spectra via masked patch reconstruction. By further aligning outputs from the 3D encoder and spectrum encoder using a contrastive objective, we enhance the 3D encoder's understanding of molecules. Evaluations on public benchmarks reveal that our pre-trained representations surpass existing methods in predicting molecular properties and modeling dynamics.

## 1 Introduction

Learning 3D molecular representations from geometric conformations offers a promising approach for understanding molecular geometry and predicting quantum properties and interactions, which is significant in drug discovery and materials science (Musaelian et al., 2023; Batatia et al., 2022; Liao & Smidt, 2023; Wang et al., 2023b; Du et al., 2023b). Given the scarcity of molecular property labels, self-supervised representation pre-training has been proposed and utilized to provide generalizable representations (Hu et al., 2020; Rong et al., 2020; Ma et al., 2024).

In contrast to contrastive learning (Wang et al., 2022; Kim et al., 2022) and masked modeling (Hou et al., 2022; Liu et al., 2023c; Wang et al., 2024b) on 2D molecular graphs and molecular languages (e.g., SMILES), the design of pre-training strategies on 3D molecular geometries is more closely aligned with physical principles. Previous studies (Zaidi et al., 2023; Jiao et al., 2023) have guided representation learning through denoising processes on 3D molecular geometries, theoretically demonstrating that denoising 3D geometries is equivalent to learning molecular force fields, specifically the negative gradient of molecular potential energy with respect to position. Essentially, these studies reveal that *establishing the relationship between 3D geometries and the energy states of molecular systems is an effective pathway to learn 3D molecular representations*.

However, existing methods are limited to the continuous description (i.e., the potential energy function) of the molecular energy states within the classical mechanics, overlooking the quantized (discrete) energy level structures from the quantum mechanical perspective. From the quantum perspective, molecular systems exhibit quantized energy level structures, meaning that energy states

---

*Correspondence to Liang Wang: `liang.wang@cripac.ia.ac.cn`
†Corresponding authors: Yu Rong and Qiang Liu

[1]The code is released at `https://github.com/AzureLeon1/MolSpectra`

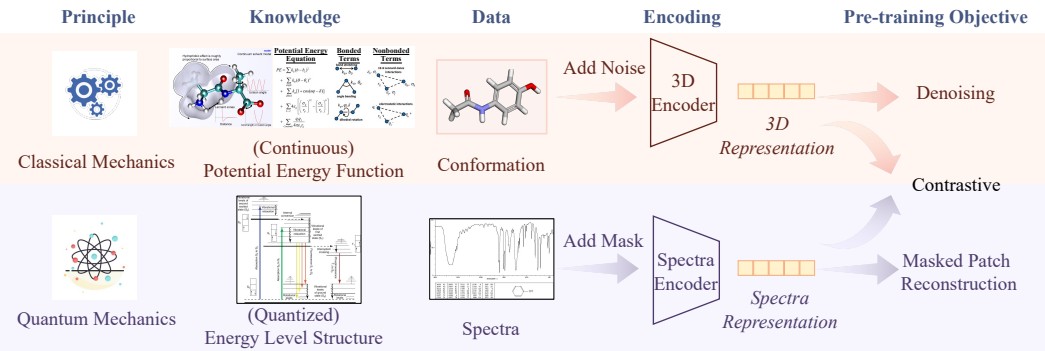

Figure 1: The conceptual view of MolSpectra, which leverages both molecular conformation and spectra for pre-training. Prior works only model classical mechanics by denoising on conformations.

can only assume specific discrete values. Specifically, different types of molecular motion, such as electronic, vibrational, and rotational motion, correspond to different energy level structures. Knowledge of these energy levels is crucial in molecular physics and quantum chemistry, as they determine the spectroscopic characteristics, chemical reactivity, and many other important molecular properties. Fortunately, experimental measurements of molecular energy spectra can reflect these structures. Meanwhile, there are many molecular spectra data obtained through experimental measurements or simulations (Zou et al., 2023; Alberts et al., 2024). Therefore, *incorporating the knowledge of energy levels into molecular representation learning is expected to facilitate the development of more informative molecular representations.*

In this paper, we propose MolSpectra, a framework that incorporates molecular spectra into the pre-training of 3D molecular representations, thereby infusing the knowledge of quantized energy level structures into the representations, as shown in Figure 1. In MolSpectra, we introduce a multi-spectrum encoder, SpecFormer, to capture both intra-spectrum and inter-spectrum peak correlations by training with a masked patches reconstruction (MPR) objective. Additionally, we employ a contrastive objective to distills the spectral features and its inherent knowledge into the learning of 3D representations. After pre-training, the resulting 3D encoder can be fine-tuned for downstream tasks, providing expressive 3D molecular representations without the need for associated spectral data. Extensive experiments over different downstream molecular property prediction benchmarks shows the superiority of MolSpectra.

In summary, our contributions are as follows:

- We introduce quantized energy level structures and molecular spectra into 3D molecular representation pre-training for the first time, surpassing previous work that relied solely on physical knowledge within the scope of classical mechanics.

- We propose SpecFormer as an expressive multi-spectrum encoder, along with the masked patches reconstruction objective for spectral representation learning.

- We propose a contrastive objective to align molecular representations in the 3D modality and spectral modalities, enabling the pre-trained 3D encoder to infer molecular spectral features in downstream tasks without relying on spectral data.

- Experiments across different downstream benchmarks demonstrate that our method effectively enhances the expressiveness of the pre-trained 3D molecular representations.

## 2 PRELIMINARIES

### 2.1 NOTATIONS

Consider a molecule characterized by its 3D structure and spectra, represented as $\mathcal{M} = (\boldsymbol{a}, \boldsymbol{x}, \mathcal{S})$. Here, $\boldsymbol{a} \in \{1, 2, \ldots, 118\}^N$ specifies the atomic numbers, indicating the types of atoms within the molecule. The vector $\boldsymbol{x} \in \mathbb{R}^{3N}$ describes the conformation of the molecule, while $\mathcal{S}$ represents its

spectra. The parameter $N$ denotes the number of atoms in the molecule. Note that the atoms are arranged in the same order in both $a$ and $x$, ensuring consistency between the atomic numbers and their corresponding spatial coordinates.

$\mathcal{S} = (s_1, \ldots, s_{|\mathcal{S}|})$ represents the set of spectra for a molecule, where $|\mathcal{S}|$ denotes the number of spectrum types considered. In our study, we focus on three types, so $|\mathcal{S}| = 3$. The first spectrum, $s_1 \in \mathbb{R}^{601}$, is the UV-Vis spectrum, which spans from 1.5 to 13.5 eV with 601 data points at intervals of 0.02 eV. The second spectrum, $s_2 \in \mathbb{R}^{3501}$, is the IR spectrum, covering a range from 500 to 4000 cm$^{-1}$ with 3501 data points at intervals of 1 cm$^{-1}$. The third spectrum, $s_3 \in \mathbb{R}^{3501}$, is the Raman spectrum, with the same range and intervals as the IR spectrum. Together, these spectra provide a comprehensive description of the molecular characteristics across different spectral modalities.

## 2.2 PRE-TRAINING 3D MOLECULAR REPRESENTATION VIA DENOISING

Denoising has emerged as a prominent pre-training objective in 3D molecular representation learning, excelling in various downstream tasks. This method involves training models to predict and remove noise introduced deliberately into molecular structures. This approach is physically interpretable due to its proven equivalence to learning the molecular force field.

**Equivalence between denoising and learning molecular force fields.** The equivalence between coordinate denoising and force field learning is established by Zaidi et al. (2023). For a given molecule $\mathcal{M}$, perturb its equilibrium structure $x_0$ according to the distribution $p(x|x_0)$, where $x$ is the noisy conformation. Assuming the molecular distribution adheres to the energy-based Boltzmann distribution with respect to the energy function $E(\cdot)$, then

$$\mathcal{L}_{\text{Denoising}}(\mathcal{M}) = \mathbb{E}_{p(x|x_0)p(x_0)} \|\text{GNN}_\theta(x) - (x - x_0)\|^2$$
$$\simeq \mathbb{E}_{p(x)} \|\text{GNN}_\theta(x) - (-\nabla_x E(x))\|^2, \tag{1}$$

where $\text{GNN}_\theta(x)$ denotes a graph neural network parameterized by $\theta$, which processes the conformation $x$ to produce node-level predictions. The notation $\simeq$ signifies the equivalence of different objectives. The proof of this equivalence is provided in the Appendix A. In prior research, the energy function $E(\cdot)$ has been defined in several forms. Below are three representative studies.

**Energy function I: mixture of isotropic Gaussians.** In Coord (Zaidi et al., 2023), the energy function is approximated using a mixture of isotropic Gaussians centered at the known equilibrium structures to replace the Boltzmann distribution, since these structures are local maxima of the Boltzman distribution. Leveraging the equivalence between the score-matching objective and denoising autoencoders (Vincent, 2011), the following denoising-based energy function $E_{\text{Coord}}(\cdot)$ is derived:

$$E_{\text{Coord}}(x) = \frac{1}{2\tau_c^2}(x - x_0)^\top (x - x_0). \tag{2}$$

Note that this objective is derived under the assumption of isotropic Gaussian noise, i.e., $p(x|x_0) \sim \mathcal{N}(x_0, \tau_c^2 I_{3N})$, where $I_{3N}$ represents the identity matrix of size $3N$, and the subscript $c$ indicates the coordinate denoising approach.

**Energy function II: mixture of anisotropic Gaussians.** Considering rigid and flexible components in molecular structures, isotropic Gaussian can lead to significant approximation errors. To address the anisotropic distribution, Frad (Feng et al., 2023) introduces hybrid noise on dihedral angles of rotatable bonds and atomic coordinates, incorporating fractional denoising of the coordinate noise. The equilibrium structure $x_0$ is initially perturbed by dihedral angle noise $p(\psi_a|\psi_0) \sim \mathcal{N}(\psi_0, \sigma_f^2 I_m)$, followed by coordinate noise $p(x|x_a) \sim \mathcal{N}(x_a, \tau_f^2 I_{3N})$. Here, $\psi_a, \psi_0 \in [0, 2\pi)^m$ represent to the dihedral angles of rotatable bonds in structures $x_a$ and $x_0$, respectively, with $m$ denoting the number of rotatable bonds. The subscript $f$ indicates the fractional denoising approach. Subsequently, the energy function is induced:

$$E_{\text{Frad}}(x) \approx \frac{1}{2}(x - x_0)^\top \Sigma_{\tau_f, \sigma_f}^{-1}(x - x_0), \tag{3}$$

where $\Sigma_{\tau_f, \sigma_f} = \tau_f^2 I_{3N} + \sigma_f^2 CC^\top$, and $C \in \mathbb{R}^{3N \times m}$ is a matrix used to linearly transform the dihedral angle noise into coordinate change, expressed as $\Delta x \approx C\Delta\psi$.

**Energy function III: classical potential energy theory.** SliDe (Ni et al., 2024) derives energy function from classical molecular potential energy theory (Alavi, 2020; Zhou & Liu, 2022). In this

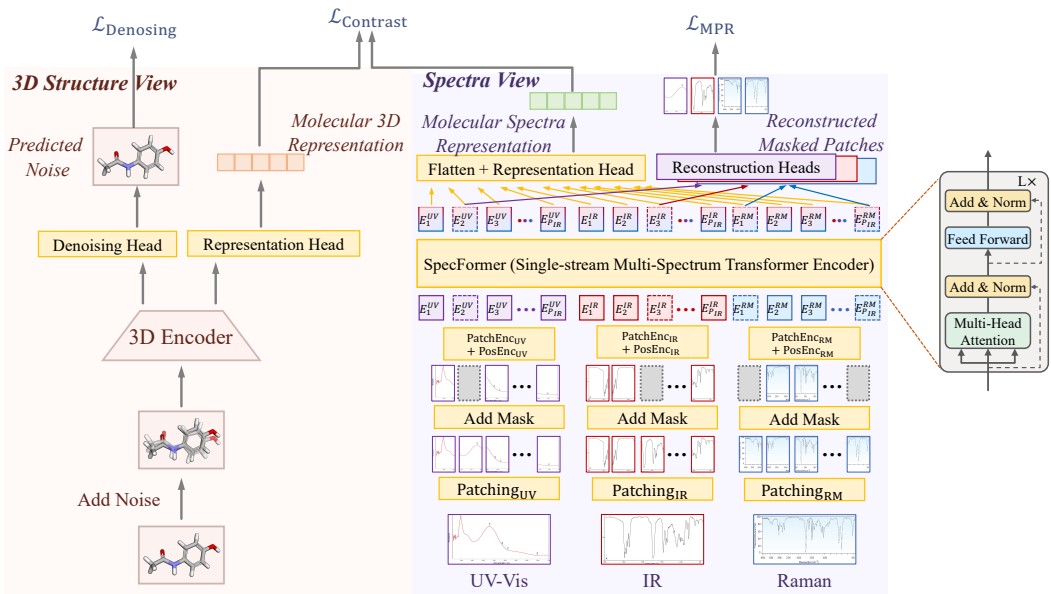

Figure 2: Overview of the MolSpectra pre-training framework. Our pre-training framework comprises three sub-objectives: the denoising objective and the MPR objective, which respectively guide the representation learning of the 3D and spectral modalities, and the contrastive objective, which aligns the representations of both modalities.

form, the total intramolecular potential energy is mainly attributed to three types of interactions: bond stretching, bond angle bending, and bond torsion. The following energy function is derived:

$$
\begin{aligned}
E_{\text{SliDe}}(\boldsymbol{r}, \boldsymbol{\theta}, \boldsymbol{\phi}) =& \frac{1}{2}[\boldsymbol{k}^B \odot (\boldsymbol{r} - \boldsymbol{r}_0)]^\top (\boldsymbol{r} - \boldsymbol{r}_0) + \frac{1}{2}[\boldsymbol{k}^A \odot (\boldsymbol{\theta} - \boldsymbol{\theta}_0)]^\top (\boldsymbol{\theta} - \boldsymbol{\theta}_0) \\
&+ \frac{1}{2}[\boldsymbol{k}^T \odot (\boldsymbol{\phi} - \boldsymbol{\phi}_0)]^\top (\boldsymbol{\phi} - \boldsymbol{\phi}_0),
\end{aligned}
\tag{4}
$$

where $\boldsymbol{r} \in (\mathbb{R}_{\geq 0})^{m_1}, \boldsymbol{\theta} \in [0, 2\pi)^{m_2}, \boldsymbol{\phi} \in [0, 2\pi)^{m_3}$ represent vectors of the bond lengths, bond angles, and bond torsion angles of the molecule, respectively. $\boldsymbol{r}_0, \boldsymbol{\theta}_0, \boldsymbol{\phi}_0$ correspond to the respective equilibrium values. The parameter vectors $\boldsymbol{k}^B, \boldsymbol{k}^A, \boldsymbol{k}^T$ determine the interaction strength.

## 3 THE PROPOSED MOLSPECTRA METHOD

Considering the complementarity of different spectra, we introduce multiple spectra into molecular representation learning. To effectively comprehend molecular spectra, we designed a Transformer-based multi-spectrum encoder, SpecFormer, along with a masked reconstruction objective to guide its training. Finally, a contrastive objective is employed to align the 3D encoding guided by the denoising objective with the spectra encoding guided by the reconstruction objective, endowing the 3D encoding with the capability to understand spectra and the knowledge they encompass.

### 3.1 SPECFORMER: A SINGLE-STREAM ENCODER FOR MULTI-MODAL ENERGY SPECTRA

For different types of spectra, each spectrum is independently patched and initially encoded. Then, all the resulting patch embeddings are concatenated and encoded using a Transformer-based encoder.

**Patching.** Compared to directly encoding individual frequency points, we divided each spectrum into multiple patches. This approach offers two distinct advantages: (i) By forming patches from adjacent frequency points, local semantic features, such as absorption peaks, can be captured more effectively. (ii) It reduces the computational overhead of subsequent Transformer layers. Technically, each spectrum $\boldsymbol{s}_i \in \mathbb{R}^{L_i}$ where $i = 1, \cdots, |\mathcal{S}|$ is first divided into patches according to the patch length $P_i$ and the stride $D_i$. When $0 < D_i < P_i$, the consecutive patches will be overlapped

with overlapping region length $P_i - D_i$. When $D_i = P_i$, the consecutive patches will be non-overlapped. $L_i$ denotes the length of $s_i$. The patching process on each spectrum will generate a sequence of patches $p_i \in \mathbb{R}^{N_i \times P_i}$, where $N_i = \left\lfloor \frac{L_i - P_i}{D_i} \right\rfloor + 1$ is the number of patches.

**Patch encoding and position encoding.** Prior to be fed into the encoder, the patches of the $i$-th spectrum are mapped to the latent space of dimension $d$ via a trainable linear projection $\boldsymbol{W}_i \in \mathbb{R}^{P_i \times d}$. A learnable additive position encoding $\boldsymbol{W}_i^{\text{pos}} \in \mathbb{R}^{N_i \times d}$ is applied to maintain the order of the patches: $\boldsymbol{p}_i' = \boldsymbol{p}_i \boldsymbol{W}_i + \boldsymbol{W}_i^{\text{pos}}$, where $\boldsymbol{p}_i' \in \mathbb{R}^{N_i \times d}$ denotes the latent representation of the spectrum $s_i$ that will be fed into the subsequent SpecFormer encoder.

**SpecFormer: multi-spectrum Transformer encoder.** Although several encoders have been proposed to map molecular spectrum into implicit representations, such as the CNN-AM (Tao et al., 2024) based on one-dimensional convolution, these encoders are designed to encode only a single type of spectrum. In our approach, multiple molecular spectra (UV-Vis, IR, Raman) are jointly considered. When encoding multiple spectra of a molecule simultaneously, an observation caught our attention and led us to adopt a Transformer-based encoder with multiple spectra as input, similar to the single-stream Transformer in multi-modal learning (Shin et al., 2021).

The observation refers to the fact that the same functional group not only causes multiple peaks within a single spectrum, but also generates peaks across different spectra. As shown on the left of Figure 3, the different vibrational modes of the methyl group (-CH$_3$) in methanol (CH$_3$OH) result in three peaks in the IR spectrum, indicating *intra-spectrum dependencies* among these peaks. A similar phenomenon occurs with the hydroxyl group (-OH) in methanol. Additionally, the aromatic ring in phenol (C$_6$H$_5$OH), shown on the right of

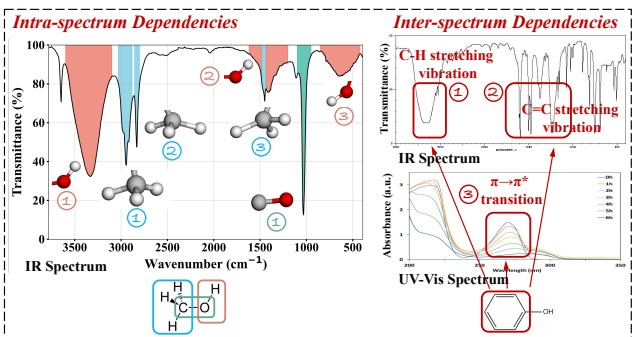

Figure 3: Illustration of intra-spectrum (left) and inter-spectrum (right) dependencies.

Figure 3, not only produces multiple peaks in the IR spectrum due to different vibrational modes but also causes an absorption peak near 270 nm in the UV-Vis spectrum due to the $\pi \to \pi^*$ transition in the aromatic ring, demonstrating the existence of *inter-spectrum dependencies*. Such dependencies have been theoretically studied, for example, in the context of vibronic coupling (Kong et al., 2021).

To capture intra-spectrum and inter-spectrum dependencies, we concatenate the embeddings obtained from patch encoding and position encoding of different spectra: $\hat{\boldsymbol{p}} = \boldsymbol{p}_1' \| \cdots \| \boldsymbol{p}_{|\mathcal{S}|}' \in \mathbb{R}^{(\sum_{i=1}^{|\mathcal{S}|} N_i) \times d}$, and then input them into the Transformer encoder as depicted in Figure 2. Then each head $h = 1, \ldots, H$ in multi-head attention will transform them into query matrices $\boldsymbol{Q}_h = \hat{\boldsymbol{p}} \boldsymbol{W}_h^Q$, key matrices $\boldsymbol{K}_h = \hat{\boldsymbol{p}} \boldsymbol{W}_h^K$ and value matrices $\boldsymbol{V}_h = \hat{\boldsymbol{p}} \boldsymbol{W}_h^V$, where $\boldsymbol{W}_h^Q, \boldsymbol{W}_h^K \in \mathbb{R}^{d \times d_k}$ and $\boldsymbol{W}_h^V \in \mathbb{R}^{d \times \frac{d}{H}}$. Afterward, a scaled product is utilized to obtain the attention output $\boldsymbol{O}_h \in \mathbb{R}^{(\sum_{i=1}^{|\mathcal{S}|} N_i) \times \frac{d}{H}}$:

$$\boldsymbol{O}_h = \text{Attention}(\boldsymbol{Q}_h, \boldsymbol{K}_h, \boldsymbol{V}_h) = \text{Softmax}\left(\frac{\boldsymbol{Q}_h \boldsymbol{K}_h^\top}{\sqrt{d_k}}\right) \boldsymbol{V}_h. \tag{5}$$

The multi-head attention block also includes BatchNorm layers and a feed forward network with residual connections as shown in Figure 2. After combining the outputs of all heads, it generates the representation denoted as $\boldsymbol{z} \in \mathbb{R}^{(\sum_{i=1}^{|\mathcal{S}|} N_i) \times d}$. Finally, a flatten layer with representation projection head is used to obtain the molecular spectra representation $\boldsymbol{z}_s \in \mathbb{R}^d$.

### 3.2 MASKED PATCHES RECONSTRUCTION PRE-TRAINING FOR SPECTRA

Before distilling the spectra information into 3D molecular representation learning, we need first ensure that the spectrum encoder can effectively comprehend molecular spectra and generate spectral representations. Considering the success of masking modeling across various domains (Devlin et al., 2019; He et al., 2022; Hou et al., 2022; Xia et al., 2023; Wang et al., 2024b; Nie et al., 2023), we propose a masked patches reconstruction (MPR) objective to guide the training of SpecFormer.

After the patching step, we randomly select a portion of patches according to the mask ratio $\alpha$ and replace them with zero vectors to implement the masking. Subsequently, the masked patches undergo patch encoding and position encoding. In this way, the semantics of the masked patches (the absorption intensity at specific wavelengths) are obscured during patch encoding, while the positional information is retained to facilitate the reconstruction of the original semantics.

After encoding by SpecFormer, the encoded results corresponding to the masked patches are input into a spectrum-specific reconstruction head to reconstruct the original spectral values that were masked. The mean squared error (MSE) between the reconstruction results and the original masked spectra serves as the loss function for the MPR task, guiding the training of SpecFormer:

$$\mathcal{L}_{\text{MPR}} = \sum_{i=1}^{|\mathcal{S}|} \mathbb{E}_{p_{i,j} \in \widetilde{\mathcal{P}}_i} \|\hat{\boldsymbol{p}}_{i,j} - \boldsymbol{p}_{i,j}\|_2^2, \tag{6}$$

where $\widetilde{\mathcal{P}}_i$ denotes the set of masked patches in the $i$-th type of molecular spectra, and $\hat{\boldsymbol{p}}_{i,j}$ denotes the reconstructed patch corresponding to the masked patch $\boldsymbol{p}_{i,j}$.

### 3.3 CONTRASTIVE LEARNING BETWEEN 3D STRUCTURES AND SPECTRA

Under the guidance of the denoising objective for 3D representation learning and the MPR objective for spectral representation learning, we further introduce a contrastive objective to align the representations across these two modalities. We treat the 3D representation $\boldsymbol{z}_x \in \mathbb{R}^d$ and spectral representation $\boldsymbol{z}_s \in \mathbb{R}^d$ of the same molecule as positive samples, and negative samples otherwise. Subsequently, the consistency between positive samples and the discrepancy between negative samples are maximized through the contrastive objective. Given the theoretical and empirical effectiveness, we employ InfoNCE (van den Oord et al., 2018) as the contrastive objective:

$$\mathcal{L}_{\text{Contrast}} = -\frac{1}{2}\mathbb{E}_{p(\boldsymbol{z}_x, \boldsymbol{z}_s)}\left[\log \frac{\exp(f_x(\boldsymbol{z}_x, \boldsymbol{z}_s))}{\exp(f_x(\boldsymbol{z}_x, \boldsymbol{z}_s)) + \sum_j \exp(f_x(\boldsymbol{z}_x^j, \boldsymbol{z}_s))} + \log \frac{\exp(f_s(\boldsymbol{z}_s, \boldsymbol{z}_x))}{\exp(f_s(\boldsymbol{z}_s, \boldsymbol{z}_x)) + \sum_j \exp(f_s(\boldsymbol{z}_s^j, \boldsymbol{z}_x))}\right], \tag{7}$$

where $\boldsymbol{z}_x^j, \boldsymbol{z}_s^j$ are randomly sampled 3D and spectra views regarding to the positive pair $(\boldsymbol{z}_x, \boldsymbol{z}_s)$. $f_x(\boldsymbol{z}_x, \boldsymbol{z}_s)$ and $f_s(\boldsymbol{z}_s, \boldsymbol{z}_x)$ are scoring functions for the two corresponding views, with flexible formulations. Here we adopt $f_x(\boldsymbol{z}_x, \boldsymbol{z}_s) = f_s(\boldsymbol{z}_s, \boldsymbol{z}_x) = \langle \boldsymbol{z}_x, \boldsymbol{z}_s \rangle$.

Note that the denoising objective can utilize any form from existing 3D molecular representation pre-training studies, enabling seamless integration of our method into these frameworks.

### 3.4 TWO-STAGE PRE-TRAINING PIPELINE

Previous pre-training efforts for 3D molecular representation have been conducted on unlabeled datasets using denoising objective. These datasets typically provide only equilibrium 3D structures without offering spectra for all molecules. To enhance the pre-training effect by incorporating spectra while leveraging denoising pre-training, we employ a two-stage pre-training approach. The first stage involves training on a larger dataset (Nakata & Shimazaki, 2017) without spectra using only the denoising objective. Subsequently, the second stage involves training on a dataset that includes spectra using the complete objective as follows:

$$\mathcal{L} = \beta_{\text{Denoising}}\mathcal{L}_{\text{Denoising}} + \beta_{\text{MPR}}\mathcal{L}_{\text{MPR}} + \beta_{\text{Contrast}}\mathcal{L}_{\text{Contrast}}, \tag{8}$$

where $\beta_{\text{Denoising}}$, $\beta_{\text{MPR}}$, and $\beta_{\text{Contrast}}$ denote the weights of each sub-objective.

## 4 EXPERIMENTS

To comprehensively evaluate the impact of molecular spectra on molecular tasks, we first verify the effectiveness of molecular spectra in the training-from-scratch method for the downstream task. Furthermore, we evaluate the effectiveness of our pre-training framework MolSpectra.

### 4.1 EFFECTIVENESS OF MOLECULAR SPECTRA IN TRAINING FROM SCRATCH

This pilot experiment aims to demonstrate the rationality for incorporating molecular spectra into pre-training. We introduce additional spectral features into a train-from-scratch molecular property

prediction model to observe the impact of spectral information on prediction outcomes. We employ EGNN (Satorras et al., 2021), a representative 3D molecular encoder, equipped with an MLP-based prediction head as the baseline model. While EGNN encodes the 3D representations, the UV-Vis spectrum of each molecule provided by the QM9S (Zou et al., 2023) dataset is encoded into spectral representations by a spectrum encoder. Before making predictions with the final MLP, we concatenate the spectral and 3D representations for prediction. The results are presented in Table 1.

Table 1: Performance (MAE ↓) when training from scratch on QM9 dataset.

| Task | $\mu$ | $\alpha$ | homo | lumo | gap | $R^2$ | ZPVE | $U_0$ | $U$ | $H$ | $G$ | $C_v$ |
| Units | (D) | $(a_0^3)$ | (meV) | (meV) | (meV) | $(a_0^2)$ | (meV) | (meV) | (meV) | (meV) | (meV) | $(\frac{cal}{mol \cdot K})$ |
| --- | --- | --- | --- | --- | --- | --- | --- | --- | --- | --- | --- | --- |
| w/o spectra | 0.029 | 0.071 | 29 | 25 | 48 | 0.106 | 1.55 | 11 | 12 | 12 | 12 | 0.031 |
| w/ spectra | **0.027** | **0.049** | **28** | **24** | **43** | **0.084** | **1.45** | **10** | **11** | **10** | **10** | **0.030** |

We observe that by directly concatenating spectral representations, the performance of molecular property prediction can be effectively enhanced. This indicates that the information from molecular spectra is beneficial for downstream molecular property prediction. Further incorporating molecular spectra into the pre-training phase of molecular representation has the potential to enhance the informativeness and generalization capability of the representations, thereby broadly improving the performance of downstream tasks.

## 4.2 EFFECTIVENESS OF MOLECULAR SPECTRA IN REPRESENTATION PRE-TRAINING

We conduct experiments to evaluate MolSpectra by first introducing spectral data into the pre-training of 3D representations, followed by evaluating the performance on downstream tasks. For a comprehensive comparison, two types of baselines are adopted: (1) training-from-scratch methods, including SchNet (Schütt et al., 2017), EGNN, DimeNet (Klicpera et al., 2020b), DimeNet++ (Klicpera et al., 2020a), PaiNN (Schütt et al., 2021), SphereNet (Liu et al., 2021), and TorchMD-Net (Thölke & Fabritiis, 2022); and (2) pre-training methods, including Transformer-M (Luo et al., 2023), SE(3)-DDM (Liu et al., 2023b), 3D-EMGP (Jiao et al., 2023), and Coord.

MolSpectra can be seamlessly plugged into any existing denoising method. To evaluate the enhancement provided by our method compared to denoising alone, we select the representative coordinate denoising (Coord) as our denoising sub-objective. This method also serves as our primary baseline.

### 4.2.1 PRE-TRAINING DATASET.

As described in Section 3.4, we first perform denoising pre-training on the PCQM4Mv2 (Nakata & Shimazaki, 2017) dataset, followed by a second stage of pre-training on the QM9Spectra (QM9S) (Zou et al., 2023) dataset, which includes multi-modal molecular energy spectra. In both stages, we adopt the denoising objective provided by Coord (Zaidi et al., 2023), as defined in Eq. 2.

The QM9S dataset comprises organic molecules from the QM9 (Ramakrishnan et al., 2014) dataset. The UV-Vis, IR, and Raman spectra of the molecules are calculated at the B3LYP/def-TZVP level of theory, through frequency analysis and time-dependent density functional theory (TD-DFT).

### 4.2.2 QM9

The QM9 dataset is a quantum chemistry dataset comprising over 134,000 small molecules, each consisting of up to 9 heavy atoms (C, N, O, F) and additional H atoms. This dataset provides an equilibrium geometric conformation for each molecule along with 12 property labels. The dataset is divided into a training set of 110k molecules, a validation set of 10k molecules, and a test set containing the remaining over 10k molecules. Prediction errors are measured using the mean absolute error (MAE). The experimental results are presented in Table 2.

The 3D molecular representations pre-trained using our method are fine-tuned and used for prediction across various properties, achieving state-of-the-art performance in 8 out of 12 properties and outperforms Coord in 10 out of 12 properties. In conjunction with the observations in Section 4.1, the performance improvement can be attributed to our incorporation of an understanding of molecular spectra and the knowledge they entail into the 3D molecular representations.

Table 2: Performance (MAE↓) on QM9 dataset. The compared methods are divided into two groups: training from scratch and pre-training then fine-tuning. The best results are highlighted in bold.

| | $\mu$ (D) | $\alpha$ ($a_0^3$) | homo (meV) | lumo (meV) | gap (meV) | $R^2$ ($a_0^2$) | ZPVE (meV) | $U_0$ (meV) | $U$ (meV) | $H$ (meV) | $G$ (meV) | $C_v$ ($\frac{cal}{mol \cdot K}$) |
|---|---|---|---|---|---|---|---|---|---|---|---|---|
| SchNet | 0.033 | 0.235 | 41.0 | 34.0 | 63.0 | 0.070 | 1.70 | 14.00 | 19.00 | 14.00 | 14.00 | 0.033 |
| EGNN | 0.029 | 0.071 | 29.0 | 25.0 | 48.0 | 0.106 | 1.55 | 11.00 | 12.00 | 12.00 | 12.00 | 0.031 |
| DimeNet++ | 0.030 | 0.044 | 24.6 | 19.5 | 32.6 | 0.330 | 1.21 | 6.32 | 6.28 | 6.53 | 7.56 | 0.023 |
| PaiNN | 0.012 | 0.045 | 27.6 | 20.4 | 45.7 | 0.070 | 1.28 | 5.85 | 5.83 | 5.98 | 7.35 | 0.024 |
| SphereNet | 0.025 | 0.045 | 22.8 | 18.9 | 31.1 | 0.270 | **1.12** | 6.26 | 6.36 | 6.33 | 7.78 | 0.022 |
| TorchMD-Net | 0.011 | 0.059 | 20.3 | 17.5 | 36.1 | **0.033** | 1.84 | 6.15 | 6.38 | 6.16 | 7.62 | 0.026 |
| Transformer-M | 0.037 | **0.041** | 17.5 | 16.2 | 27.4 | 0.075 | 1.18 | 9.37 | 9.41 | 9.39 | 9.63 | 0.022 |
| SE(3)-DDM | 0.015 | 0.046 | 23.5 | 19.5 | 40.2 | 0.122 | 1.31 | 6.92 | 6.99 | 7.09 | 7.65 | 0.024 |
| 3D-EMGP | 0.020 | 0.057 | 21.3 | 18.2 | 37.1 | 0.092 | 1.38 | 8.60 | 8.60 | 8.70 | 9.30 | 0.026 |
| Coord | 0.016 | 0.052 | 17.7 | 14.7 | 31.8 | 0.450 | 1.71 | 6.57 | 6.11 | 6.45 | 6.91 | **0.020** |
| MolSpectra | **0.011** | 0.048 | **15.5** | **13.1** | **26.8** | 0.410 | 1.71 | **5.67** | **5.45** | **5.87** | **6.18** | 0.021 |

Table 3: Performance (MAE↓) on MD17 force prediction (kcal/mol/ Å). The methods are divided into two groups: training from scratch and pre-training then fine-tuning. The best results are in bold.

| | Aspirin | Benzene | Ethanol | Malonal -dehyde | Naphtha -lene | Salicy -lic Acid | Toluene | Uracil |
|---|---|---|---|---|---|---|---|---|
| SphereNet | 0.430 | 0.178 | 0.208 | 0.340 | 0.178 | 0.360 | 0.155 | 0.267 |
| SchNet | 1.350 | 0.310 | 0.390 | 0.660 | 0.580 | 0.850 | 0.570 | 0.560 |
| DimeNet | 0.499 | 0.187 | 0.230 | 0.383 | 0.215 | 0.374 | 0.216 | 0.301 |
| PaiNN | 0.338 | - | 0.224 | 0.319 | 0.077 | 0.195 | 0.094 | 0.139 |
| TorchMD-Net | 0.245 | 0.219 | 0.107 | 0.167 | 0.059 | 0.128 | 0.064 | 0.089 |
| SE(3)-DDM* | 0.453 | - | 0.166 | 0.288 | 0.129 | 0.266 | 0.122 | 0.183 |
| Coord | 0.211 | 0.169 | 0.096 | 0.139 | **0.053** | 0.109 | **0.058** | **0.074** |
| MolSpectra | **0.099** | **0.097** | **0.052** | **0.077** | 0.085 | **0.093** | 0.075 | 0.095 |

### 4.2.3 MD17

The MD17 dataset contains molecular dynamics trajectories for eight organic molecules, including aspirin, benzene, and ethanol. It offers 150k to nearly 1M conformations per molecule, with energy and force labels. Unlike QM9, MD17 emphasizes dynamic behavior in addition to static properties. We use a standard limited data split: models train on 1k samples, validate on 50, and test on the rest. Performance is evaluated using MAE, with results in Table 3.

Our approach also results in the expected performance improvement on MD17. MD17 is a dataset comprising a large number of non-equilibrium molecular structures and their corresponding force fields, which serves to evaluate a model's understanding of molecular dynamics. However, previous pre-training methods based solely on denoising have only learned force field patterns at static equilibrium states, failing to adequately capture the dynamic evolution of molecular systems. In contrast, our MolSpectra learns the dynamic evolution of molecules by understanding energy level transition patterns, thereby outperforming denoising-based pre-training methods.

### 4.3 SENSITIVITY ANALYSIS OF PATCH LENGTH $P_i$, STRIDE $D_i$, AND MASK RATIO $\alpha$

We conduct experiments to evaluate the impact of patch length $P_i$, stride $D_i$, and mask ratio $\alpha$. Results are summarized in Table 4 and Table 5.

From Table 4, we observe that when consecutive patches have overlap ($D_i < P_i$), the performance of pre-training is superior compared to scenarios without overlap ($D_i = P_i$). Specifically, the performance is optimal when the stride is half of the patch length. This is because appropriate overlap can better preserve and capture local features, particularly the information at the patch boundaries. Additionally, we find that choosing an appropriate patch length further enhances performance. In our experiments, the configuration of $P_i = 20, D_i = 10$ yields the best results.

Table 4: Sensitivity of patch length and stride.

| patch length | stride | overlap ratio | homo | lumo | gap |
|---|---|---|---|---|---|
| 20 | 5 | 75% | 15.9 | 13.7 | 28.0 |
| 20 | 10 | 50% | **15.5** | **13.1** | **26.8** |
| 20 | 15 | 25% | 16.1 | 13.6 | 28.1 |
| 20 | 20 | 0% | 15.7 | 13.5 | 27.5 |
| 16 | 8 | 50% | 16.0 | 13.4 | 27.6 |
| 30 | 15 | 50% | 15.9 | 14.0 | 28.1 |

Table 5: Sensitivity of mask ratio.

| mask ratio | homo | lumo | gap |
|---|---|---|---|
| 0.05 | 15.7 | 13.4 | 29.7 |
| 0.10 | **15.5** | **13.1** | **26.8** |
| 0.15 | 15.7 | 13.5 | 28.0 |
| 0.20 | 16.0 | 13.6 | 28.1 |
| 0.25 | 16.3 | 13.5 | 28.0 |
| 0.30 | 16.2 | 13.7 | 29.0 |

Regarding the mask ratio, $\alpha = 0.10$ is a preferable choice. A small mask ratio results in insufficient MPR optimization, hindering SpecFormer training. Conversely, a large mask ratio causes excessive spectral perturbation, degrading performance when aligning with the 3D representations with the contrastive objective. An appropriate mask ratio strikes a balance between these two aspects.

## 4.4 ABLATION STUDY

To rigorously demonstrate the contributions of masked patches reconstruction, the incorporation of molecular spectra, and each spectral modality, we conducted an ablation study on them.

**Ablation study of masked patches reconstruction.** We remove the MPR loss to analyze the impact of masked patches reconstruction, referred to as "w/o MPR" in Table 6. Removing the MPR objective leads to performance deterioration. This is consistent with the sensitivity analysis of the mask ratio $\alpha$ in Section 4.3, as removing MPR is an extreme case where $\alpha = 0$. This decline is due to the lack of effective guidance in training SpecFormer.

Table 6: Ablation of optimization objectives.

| | homo | lumo | gap |
|---|---|---|---|
| MolSpectra | 15.5 | 13.1 | 26.8 |
| w/o MPR | 16.4 | 14.1 | 29.7 |
| w/o MPR, Contrast | 17.5 | 14.4 | 31.2 |

Using an undertrained SpecFormer for contrastive learning with 3D encoder outputs limits performance improvement.

**Ablation study of molecular spectra.** We retain only the denoising loss, removing both the MPR loss and contrastive loss, referred to as "w/o MPR, Contrast" in Table 6. The only difference between this variant and MolSpectra is the incorporation of molecular spectra into the pre-training. The "w/o MPR, Contrast" results are inferior to those of MolSpectra, highlighting that incorporating molecular spectra effectively enhances the quality and generalizability of molecular 3D representations.

**Ablation study of each spectral modality.** To evaluate the contributions of each spectral modality to the performance, we conduct an ablation study for each modality. The results are presented in Table 7. It can be observed that each spectral modality contributes differently, with the UV-Vis spectrum having the smallest contribution and the IR spectrum the largest, likely due to the varying information content in each modality.

Table 7: Ablation of spectral modalities.

| UV-Vis | IR | Raman | homo | lumo | gap |
|---|---|---|---|---|---|
| ✓ | ✓ | ✓ | 15.5 | 13.1 | 26.8 |
| - | ✓ | ✓ | 15.8 | 13.3 | 27.1 |
| ✓ | - | ✓ | 16.6 | 14.1 | 28.9 |
| ✓ | ✓ | - | 16.1 | 13.9 | 28.3 |

## 5 RELATED WORK

**3D molecular pre-training.** Molecular 2D structures are typically represented as graphs and modeled using graph learning methods (Gilmer et al., 2017; Li et al., 2023; Jiang et al., 2024; Yuan et al., 2025). However, 3D molecular structures provide critical geometric information that is essential for understanding physicochemical properties (Chen et al., 2023; 2024; Wang et al., 2024a; Sun et al., 2024; Han et al., 2024), which cannot be directly inferred from 2D graphs or SMILES representations (Gong et al., 2024). Designing effective strategies for pre-training 3D molecular representations remains challenging due to the geometric symmetries inherent in 3D structures and their strong connection to physical knowledge, such as potential energy functions.

Denoising the geometric structure has been demonstrated as an effective strategy for 3D representation pre-training (Liu et al., 2023b; Jiao et al., 2023; Kim et al., 2023; Zhou et al., 2023; Wang et al., 2025). Coordinate denoising (Coord) (Zaidi et al., 2023) first theoretically proves that the denoising objective is equivalent to learning the gradient of the potential energy with respect to positions, essentially the force field. Building on this work, fractional denoising (Frad) (Feng et al., 2023) introduces dihedral angle noise to optimize the sampling of low-energy structures. Further, SliDe (Ni et al., 2024) incorporates a more rigorous potential energy from classical mechanics. Another line of research simultaneously leverages both 2D and 3D structures for pre-training molecular representations, addressing the complementarity of the two modalities (Li et al., 2022; Zhu et al., 2022; Liu et al., 2023a; Du et al., 2023a; Yu et al., 2024) or the computational complexity of 3D structure determination (Liu et al., 2022; Stärk et al., 2022; Wang et al., 2023a).

Although these studies elucidate the relationship between molecular 3D structures and their energy states, they remain limited to the description of molecular energy states within classical mechanics, without considering the quantized energy level structures as described by quantum mechanics.

**Molecular spectroscopy.** Molecular spectroscopy studies interactions between molecules and electromagnetic radiation. Analyzing spectra provides valuable insights into molecular structure, composition, and dynamics (Lancaster et al., 2024). When encountering unknown substances, researchers conduct spectroscopic measurements on samples and compare the observed spectra with libraries for identification. To expand library coverage, machine learning methods are widely used to predict molecules' spectra (Zou et al., 2023; Wei et al., 2018; Zong et al., 2024).

Some studies incorporate physical principles into spectra prediction models as inductive biases, including molecular dynamics simulations via equivariant message passing (Schütt et al., 2021), fragmentation (Dührkop et al., 2020; Cao et al., 2020; Goldman et al., 2023a), motifs (Park et al., 2023), and long-distance atomic interactions (Young et al., 2024). Another line of research approach bypasses spectral library comparison and directly performs de novo structure elucidation from spectra (Stravs et al., 2021; Goldman et al., 2023b; Tao et al., 2024).

Since different spectroscopic techniques offer complementary advantages, the joint analysis of multiple spectra can provide comprehensive information (Alberts et al., 2024). In this study, we encodes multiple spectra, and introduce them into molecular representation pre-training for the first time.

## 6 CONCLUSION

In this study, we explore pre-training molecular 3D representations beyond classical mechanics. By leveraging the correlation between molecular energy level structures and molecular spectra in quantum mechanics, we introduce molecular spectra for pre-training molecular 3D representations (MolSpectra). By aligning the 3D encoder trained with a denoising objective and the spectrum encoder trained with a masked patch reconstruction objective, we enhance the informativeness and transferability of the resulting 3D representations.

ACKNOWLEDGMENTS

This work is jointly supported by National Science and Technology Major Project (2023ZD0120901) and National Natural Science Foundation of China (62372454, 62236010).

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

# Appendix

**Contents of the appendix**

## A  PROOF OF THEORETICAL RESULTS

**Theorem A.1** (Equivalence between the denoising objective and the learning of molecular force fields (Zaidi et al., 2023)). *Assume the conformation distribution is a mixture of Gaussian distribution centered at the equilibriums:*

$$p(\boldsymbol{x}) = \int p(\boldsymbol{x}|\boldsymbol{x}_0)p(\boldsymbol{x}_0), \ p(\boldsymbol{x}|\boldsymbol{x}_0) \sim \mathcal{N}(\boldsymbol{x}_0, \tau^2 I_{3N}) \tag{A1}$$

$\boldsymbol{x}_0, \ \boldsymbol{x} \in \mathbb{R}^{3N}$ *are equilibrium and noisy conformation respectively, $N$ is the number of atoms in the molecule. It relates to molecular energy by Boltzmann distribution $p(\boldsymbol{x}) \propto exp(-E(\boldsymbol{x}))$.*

*Then given a sampled molecule $\mathcal{M}$, the denoising loss on the conformation coordinates is an equivalent optimization target to force field prediction:*

$$\mathcal{L}_{\text{Denoising}}(\mathcal{M}) = \mathbb{E}_{p(\boldsymbol{x}|\boldsymbol{x}_0)p(\boldsymbol{x}_0)}||\text{GNN}_\theta(\boldsymbol{x}) - (\boldsymbol{x} - \boldsymbol{x}_0)||^2 \tag{A2}$$

$$\simeq \mathbb{E}_{p(\boldsymbol{x})}||\text{GNN}_\theta(\boldsymbol{x}) - (-\nabla_{\boldsymbol{x}}E(\boldsymbol{x}))||^2, \tag{A3}$$

*where $\text{GNN}_\theta(\boldsymbol{x})$ denotes a graph neural network with parameters $\theta$ which takes conformation $\boldsymbol{x}$ as an input and returns node-level noise predictions, $\simeq$ denotes equivalence.*

*Proof.* According to Boltzmann distribution, Eq. A3 is equal to $\mathbb{E}_{p(\boldsymbol{x})}||GNN_\theta(\boldsymbol{x}) - \nabla_{\boldsymbol{x}} \log p(\boldsymbol{x})||^2$. By using a conditional score matching lemma (Vincent, 2011), the equation above is further equal to $\mathbb{E}_{p(\boldsymbol{x}|\boldsymbol{x}_0)p(\boldsymbol{x}_0)}||GNN_\theta(\boldsymbol{x}) - \nabla_{\boldsymbol{x}} \log p(\boldsymbol{x}|\boldsymbol{x}_0)||^2 + T_1$, where $T_1$ is constant independent of $\theta$. Then with the Gaussian assumption, it becomes $\mathbb{E}_{p(\boldsymbol{x}|\boldsymbol{x}_0)p(\boldsymbol{x}_0)}||GNN_\theta(\boldsymbol{x}) - \frac{\boldsymbol{x}_0 - \boldsymbol{x}}{\tau_c^2}||^2 + T_1$. Finally, since coefficients $-\frac{1}{\tau^2}$ do not rely on the input $\boldsymbol{x}$, it can be absorbed into $\text{GNN}_\theta$, thus obtaining Eq. A2. $\qquad\square$

## B  VISUALIZATION AND ANALYSIS OF SPECTRA

In this section, we visualize the three types of spectra we utilize (UV-Vis, IR, Raman) and standardize the initial spectral data based on data analysis. In Figure A1, we visualize 20 randomly sampled spectra from QM9S for each type of spectrum. A notable pattern observed is that, although each spectrum consists of numerous absorption peaks, there are significant differences in the heights (absorption intensities) of these peaks. For instance, in the IR spectra, the absorption intensity at most peaks is around 200, but a few peaks reach an intensity of 800. However, in qualitative analysis, the position and shape of the peaks are more critical than their heights. Therefore, the differences in peak absorption intensities can interfere with model training under the MSE loss metric. To address this issue, we pre-process the absorption intensities of the spectra by applying a $\log_{10}$ transformation to mitigate the interference caused by peak intensity differences.

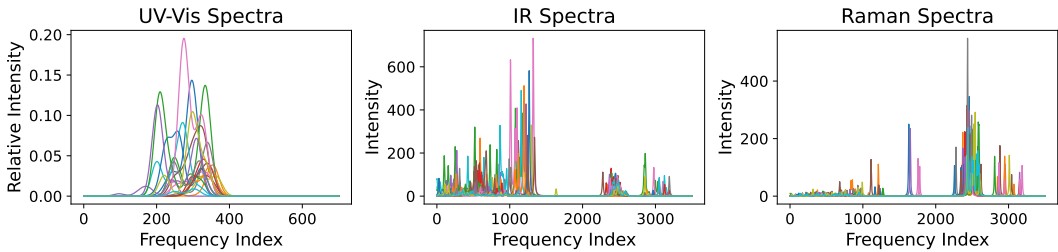

Figure A1: Randomly sampled examples of molecular energy spectra.

## C IMPLEMENTATION DETAILS

### C.1 HARDWARE AND SOFTWARE

Our experiments are conducted on Linux servers equipped with 184 Intel Xeon Platinum 8469C CPUs, 920GB RAM, and 8 NVIDIA H20 96GB GPUs. Our model is implemented in PyTorch version 2.3.1, PyTorch Geometric version 2.6.1 (https://pyg.org/) with CUDA version 12.1, and Python 3.10.14.

### C.2 MODEL CONFIGURATION

The SpecFormer is implemented using a 3-layer Transformer with 16 attention heads. Following previous works, we set both $d$ and $d_k$ as 256. TorchMD-Net (Thölke & Fabritiis, 2022) is adopted as the 3D molecular encoder. We tune the mask ratio (i.e., $\alpha$) in $\{0.05, 0.10, 0.15, 0.20, 0.25, 0.30\}$, tune the "stride/patch length" pair (i.e., $D_i/P_i$) in $\{5/20, 10/20, 15/20, 20/20, 8/16, 15/30\}$, and tune the weights of sub-objectives (i.e., $\beta_{\text{Denoising}}$, $\beta_{\text{MPR}}$, and $\beta_{\text{Contrast}}$) in $\{0.01, 0.1, 1, 10\}$. Since our goal is to align the 3D representations and spectra representations of molecules during the pre-training phase, and not rely on molecular spectra data during downstream fine-tuning, these hyper-parameters related to molecular spectra are tuned on the pre-training dataset. Based on the results of hyper-parameter tuning, we adopt $\alpha = 0.10, D_i = 10, P_i = 20, \beta_{\text{Denoising}} = 1.0, \beta_{\text{MPR}} = 1.0$, and $\beta_{\text{Contrast}} = 1.0$.

Following SimCLR (Chen et al., 2020), the contrastive loss in our Eq. 7 is implemented using in-batch contrastive loss, where positive and negative pairs are constructed within each data batch. Therefore, for each anchor representation in a batch, there is one positive sample and $bs - 1$ negative samples, where $bs$ is the batch size. In our method, $bs = 128$.

In both pre-training stages, we use the noise generation method and denoising objective provided by Coord (Zaidi et al., 2023), specifically energy function I as described in Section 2.2. The noise is added to atom positions as scaled mixture of isotropic Gaussian noise, with a scaling factor of 0.04. The denoising objective is defined in Eq. 2.

For baselines, we follow their recommended settings.

## D MORE EXPERIMENTAL RESULTS AND DISCUSSIONS

In addition to Coord, we evaluate the effect of incorporating SliDe (Ni et al., 2024) into our Mol-Spectra. SliDe is also a denoising-based pre-training method, utilizing the TorchMD-Net (Thölke & Fabritiis, 2022) as its encoder backbone, consistent with previous pre-training work (Zaidi et al., 2023; Feng et al., 2023). The results are presented in Table A1 and Table A2.

Integrating our method with SliDe effectively reduces the error in property prediction on the QM9 dataset and the MD17 dataset. Given that our method enhances both Coord and SliDe, this suggests that our approach is broadly effective across various denoising-based pretraining strategies. Furthermore, incorporating molecular spectra can guide the pre-trained model to acquire knowledge beyond what denoising objectives can offer, which proves beneficial for downstream property prediction.

Table A1: Performance (MAE↓) on QM9 dataset. The better result between the two variants of each pretraining method, w/ and w/o MolSpectra, is highlighted in bold.

| | $\mu$ (D) | $\alpha$ ($a_0^3$) | homo (meV) | lumo (meV) | gap (meV) | $R^2$ ($a_0^2$) | ZPVE (meV) |
|---|---|---|---|---|---|---|---|
| Coord | 0.016 | 0.052 | 17.7 | 14.7 | 31.8 | 0.450 | **1.71** |
| Coord w/ MolSpectra | **0.011** | **0.048** | **15.5** | **13.1** | **26.8** | **0.410** | **1.71** |
| SliDe | 0.015 | 0.050 | 18.7 | 16.2 | 28.8 | 0.606 | 1.78 |
| SliDe w/ MolSpectra | **0.012** | **0.043** | **17.0** | **15.8** | **28.5** | **0.424** | **1.73** |

Table A2: Performance (MAE↓) on MD17 dataset. The better result between the two variants of each pretraining method, w/ and w/o MolSpectra, is highlighted in bold.

| | Aspirin | Benzene | Ethanol | Malonal -dehyde | Naphtha -lene | Salicy -lic Acid | Toluene | Uracil |
|---|---|---|---|---|---|---|---|---|
| Coord | 0.211 | 0.169 | 0.096 | 0.139 | **0.053** | 0.109 | **0.058** | **0.074** |
| Coord w/ MolSpectra | **0.099** | **0.097** | **0.052** | **0.077** | 0.085 | **0.093** | 0.075 | 0.095 |
| SliDe | 0.174 | 0.169 | 0.088 | 0.154 | **0.048** | 0.101 | **0.054** | **0.083** |
| SliDe w/ MolSpectra | **0.160** | **0.054** | **0.055** | **0.088** | 0.073 | **0.098** | 0.077 | 0.097 |

# E  VISUALIZATION OF ATTENTION PATTERNS AND LEARNED SPECTRA REPRESENTATIONS IN SPECFORMER

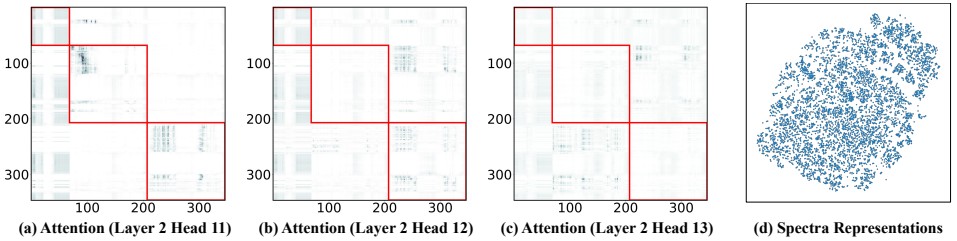

(a) Attention (Layer 2 Head 11)  (b) Attention (Layer 2 Head 12)  (c) Attention (Layer 2 Head 13)  (d) Spectra Representations

Figure A2: (a-c) Attention maps from three attention heads in SpecFormer. Different heads model distinct dependencies. (d) t-SNE visualization of the spectra representations produced by Spec-Former.

We visualize the attention patterns and learned spectra representations in SpecFormer. Based on the visualizations presented in Figure A2, we have made the following observations.

In Figure A2(a-c), we visualize attention maps from three attention heads in SpecFormer's second layer. The attention weights within the three blocks along the main diagonal indicate intra-spectrum dependencies, while those outside reveal inter-spectrum dependencies, as explained in Section 3.1. It can be observed that different attention heads model distinct dependencies: Head 11 focuses on intra-spectrum dependencies, Head 13 focuses on inter-spectrum dependencies, and Head 12 models both types simultaneously. In inter-spectrum dependencies, the interaction between IR spectra and Raman spectra is relatively pronounced, which may be related to their mutual association with vibrational modes. Additionally, because the intensity peaks and dependencies in molecular spectra are sparse, the attention maps in SpecFormer are generally sparse as well.

In Figure A2(d), we use t-SNE to visualize the spectra representations produced by the final layer of SpecFormer. It can be observed that the distribution of representations in the latent space is relatively uniform and forms several potential clusters. This well-shaped distribution of representations reveals effective spectra representation learning and supports the structure-spectrum alignment.

## F    LIMITATIONS AND POTENTIAL FUTURE DIRECTIONS

One limitation of our method is the availability, scale, and diversity of molecular spectral data. Our current dataset comprises geometric structures of 134,000 molecules, each with three types of spectra (UV-Vis, IR, Raman). To effectively explore the scaling laws of pre-training methods, larger and more diverse molecular spectral datasets are necessary. Encouragingly, molecular spectroscopy has been gaining increasing attention in the research community, with larger and more diverse datasets being released, such as the recent multimodal spectroscopic dataset (Alberts et al., 2024). This development supports advancements in molecular representation learning and other related tasks.

Another limitation is that our proposed SpecFormer can currently only handle one-dimensional molecular spectra. For higher-dimensional spectra, such as two-dimensional NMR and two-dimensional correlation spectra, further development of sophisticated spectrum encoder is needed.

Looking ahead, we envision several future directions in this field. First, there is potential in investigating the scaling laws of pre-training on larger and more diverse molecular spectral datasets. Second, expanding the scope of molecular spectrum encoding to include a wider range, such as NMR, mass spectra, and two-dimensional spectra, could be highly beneficial. Third, while a pre-trained spectral encoder has been developed in our method, we have so far only applied the pre-trained 3D encoder to downstream tasks. Exploring the use of the pre-trained spectral encoder for molecular spectrum-related downstream tasks, such as automated molecular structure elucidation from spectra, represents an promising opportunity. Finally, current molecular 3D pre-training methods are designed based on TorchMD-Net (Thölke & Fabritiis, 2022). With the development of equivariant message passing neural networks, more expressive backbone architectures, such as Allegro (Musaelian et al., 2023) and MACE (Batatia et al., 2022) have been proposed, improving the prediction of molecular properties when trained from scratch. Extending pre-training strategies to these state-of-the-art architectures holds the promise of further advancing downstream tasks.

