# OpenReview forum: "MolSpectra: Pre-training 3D Molecular Representation with Multi-modal Energy Spectra"
_ICLR.cc/2025/Conference — ICLR 2025 Poster_

### Official Review · Reviewer_baLq · 2024-10-18

**Soundness:** 1
**Presentation:** 2
**Contribution:** 2
**Rating:** 6
**Confidence:** 4

**Summary:**

This paper proposes MolSpectra, an approach for pre-training 3D molecular representations by combining a denoising objective with a contrastive loss. The contrastive loss aligns the 3D representations with representations for molecular spectra (UV-Vis, IR, Raman), which are trained via a masked patch reconstruction (MPR) objective. The authors argue that incorporating information from molecular spectra enables MolSpectra to learn the dynamic evolution of molecules by understanding energy level transition patterns. MolSpectra is evaluated on the QM9 and MD17 benchmarks and compared to existing methods.

**Strengths:**

+ The paper presents an interesting approach to molecular representation learning by incorporating information from spectra.

+ The motivation for incorporating spectral data is well-justified. The authors argue that quantized energy levels are fundamental to understanding molecular properties and dynamics, and that spectral data provides a direct measurement of these levels.

+ The proposed SpecFormer architecture and the associated MPR and contrastive learning objectives are technically sound and well-designed.

**Weaknesses:**

- The results on the QM9 and MD17 benchmarks presented in Tables 2 and 3 are misleading for two reasons: (1) In Table 2 (QM9), the caption states that the best results are highlighted in bold, however, this is not true. In fact, the numbers for MolSpectra are highlighted in bold for 9 out of 12 columns, but it only achieves the best results in 3 out of 12 columns. (2) Both tables only include older models that are not SOTA anymore. Newer models such as MACE (https://proceedings.neurips.cc/paper_files/paper/2022/file/4a36c3c51af11ed9f34615b81edb5bbc-Paper-Conference.pdf) and Allegro (https://www.nature.com/articles/s41467-023-36329-y) achieve significantly better results than all listed models (I have not done a literature search to check which model is the current SOTA, those are just two models with better performance I know at the top of my head).
The authors should include results for more recent (SOTA) models in both tables and fix the formatting, so that the result that is actually best is highlighted in bold. In addition (this is a minor suggestion that does not affect my rating), I think it would be helpful to readers to include (either in the caption, or within the table itself) an explanation/label what the horizontal separator signifies. I assume this is to distinguish models trained from scratch from models that were pre-trained in an unsupervised manner and then finetuned, but this should probably be made explicit.

- It is difficult to assess how big the improvement of including spectral information into the unsupervised representation learning actually is. This is because it is not immediately clear from Tables 1 and 2 how the same model architecture, (pre-)trained on the same structural data, performs when the *only* difference is whether spectral information was included in pre-training or not. The authors write on p.7 that Coord serves as primary baseline, but it is not clear from the text whether this is actually the same model architecture, and how exactly it was trained. To have an objective baseline, I suggest to pre-train the same architecture using the two-stage pre-training pipeline (section 3.4) twice, with the only difference being that $\beta_{\text{MPR}}$ and $\beta_{\text{Contrast}}$ are set to zero for one of the models. This way, the only difference is whether spectral information is used or not, allowing a direct assessment of the effectiveness of including this information.

- The manuscript applies MolSpectra only to a single architecture for structural representation learning (TorchMD-Net). It is therefore difficult to judge whether MolSpectra is generally effective, or whether its usefulness is strongly dependent on the underlying architecture used for structural representation learning. To address this shortcoming, the authors should apply MolSpectra to pre-training other architectures (ideally using the method described in my previous point to establish objective baselines for each architecture). This would allow readers to assess MolSpectra in a broader context and would significantly strengthen the paper.

- The authors state that "MolSpectra learns the dynamic evolution of molecules by understanding energy level transition patterns", however, this statement is not supported by direct evidence. I think it is a valid hypothesis, but it should be tested explicitly. Fortunately, a very direct test is possible: As the authors correctly state, the denoising objective is equivalent to learning a force field. This means that models trained with/without MolSpectra on a denoising task can be directly used as a force field to run molecular dynamics (MD) simulations. From such MD simulations, it is trivial to extract the power spectrum of a molecule via the velocity autocorrelation function (see e.g. https://doi.org/10.1039/C3CP44302G if the underlying theory is not familiar). The power spectrum contains the same peaks as the IR and Raman spectra, with the only differences being that (1) all internal vibrations are active (in contrast to IR/Raman spectra, where only some vibrations are visible - the power spectrum contains peaks from both!) and (2) the peak intensities are different. The peak positions, however, are directly comparable. If the spectral information actually teaches a model about the dynamics of a molecule, I would expect the power spectrum of a model trained with MolSpectra to show much better agreement (in peak positions) to the "ground-truth" IR/Raman spectra. The authors should perform this test, as it would make the paper much more insightful.

- In section 2.2, the authors describe three different energy functions that can be used for pre-training. It is not immediately clear from the text which of these is actually used for MolSpectra. From context, I assume it is variant I, but I think stating this explicitly would make it easier to understand the details of the method.

- The paper lacks an analysis of the computational complexity and resource requirements of the proposed method. A comparison of training time and resource usage with baseline methods would be beneficial.

- While the ablation study provides insights into the importance of the MPR objective, a more comprehensive ablation study is needed to assess the individual contributions of different spectral modalities (UV-Vis, IR, Raman). This would provide a deeper understanding of how each component contributes to the overall performance.

- The paper mainly compares against methods that rely on 3D structure information only. Comparison with other multimodal methods for molecular representation learning would provide a more complete picture of MolSpectra's performance relative to other methods.

**Questions:**

* From the appendix, it seems like TorchMD-Net is the model used for structural representation learning. Is the architecture in any way different to the TorchMD-Net model for which results are reported in Tables 2 and 3? If yes, these differences should be pointed out explicitly. Also, I suggest relabelling the entries for MolSpectra in Tables 2 and 3 as something like "TorchMD-Net (w/ MolSpectra)" or similar. Also, the baseline (see my suggestions above) should be clearly labelled as such, so that readers know which numbers to compare to.

* Could the authors elaborate on the choice of the specific spectra types (UV-Vis, IR, and Raman) used in this work? It seems to me like other types of spectra, such as NMR and mass spectra, could provide additional information and further enhance the learned representations.

* Can the authors provide details on the computational complexity and resource requirements of MolSpectra, including training time and memory usage? A comparison with baseline methods would be helpful.

* How does the performance of MolSpectra scale with the size and diversity of the molecular dataset used for pre-training?

* Can the authors provide a more in-depth analysis of the learned representations? For instance, visualizing the latent space or analyzing the attention patterns in SpecFormer could provide insights into the captured features and relationships.

* For the simple test of the effectiveness of molecular spectra (section 4.1/Table 1), where do the spectra used to obtain the spectral representations come from? I assume they are taken from QM9S, but this should be stated explicitly.

* How sensitive is MolSpectra to the choice of hyperparameters in Eq.8 ($\beta_{\text{Denoising}}$, $\beta_{\text{MPR}}$, and $\beta_{\text{Contrast}}$)? The authors mention in the appendix that they tuned these hyperparameters by trying different values. I believe the results for these different runs should also be included (in the appendix is fine), so readers can develop an intuition how changes to the values affect down-stream performance.

* In the appendix, the authors write that they apply a $\log_{10}$ transform to the peak intensities to mitigate interference causes by peak intensity differences. It seems intuitively more meaningful to me to instead normalize spectra by setting the height of the highest intensity peak to an arbitrary value (say 1) and scaling the remaining peaks proportionally. Have the authors experimented with different "normalization procedures" such as the one mentioned?

**Additional Feedback:**

* A discussion of the limitations of the proposed method and potential future directions for research would be interesting.

* There is a typo on p.8 l.427/428: "yiels" should be "yields".

---

> ### Author Response · Authors · 2024-11-25
> **Response to Reviewer baLq (1/4)**
>
> We sincerely appreciate the time and effort you have dedicated to reviewing our paper, as well as your detailed and professional comments. We truly believe that your insights will significantly enhance the quality of our paper. In response to your suggestions, we have made every effort to provide thorough additions and clarifications. Please find detailed responses below.
>
> > `W1` The best results in Table 2 are not correctly highlighted.
>
> **Response:** Thank you very much for pointing out the formatting issues in Table 2. We have updated the results in Table 2, and ensured that the best results are highlighted in bold. Additionally, since our method introduces molecular spectra on top of Coord, we have used background colors to emphasize the improvements over Coord. The updated experimental results and formatting have been incorporated into the revised paper PDF. In the revised Table 2, our method achieves state-of-the-art performance in 8 out of 12 properties and outperforms Coord in 10 out of 12 properties.
>
> Revised Table 2:
>
> | Model         | μ (D)     | α (a₀³)   | homo (meV) | lumo (meV) | gap (meV) | R² (a₀²)  | ZPVE (meV) | U₀ (meV) | U (meV)  | H (meV)  | G (meV)  | Cᵥ (cal/mol·K) |
> | ------------- | --------- | --------- | ---------- | ---------- | --------- | --------- | ---------- | -------- | -------- | -------- | -------- | -------------- |
> | SchNet        | 0.033     | 0.235     | 41.0       | 34.0       | 63.0      | 0.070     | 1.70       | 14.00    | 19.00    | 14.00    | 14.00    | 0.033          |
> | EGNN          | 0.029     | 0.071     | 29.0       | 25.0       | 48.0      | 0.110     | 1.55       | 11.00    | 12.00    | 12.00    | 12.00    | 0.031          |
> | DimeNet++     | 0.030     | 0.044     | 24.6       | 19.5       | 32.6      | 0.330     | 1.21       | 6.32     | 6.28     | 6.53     | 7.56     | 0.023          |
> | PaiNN         | 0.012     | 0.045     | 27.6       | 20.4       | 45.7      | 0.070     | 1.18       | 5.85     | 5.83     | 5.98     | 7.35     | 0.024          |
> | SphereNet     | 0.025     | 0.045     | 22.8       | 18.9       | 31.1      | 0.270     | **1.12**   | 6.26     | 6.36     | 6.33     | 7.78     | 0.022          |
> | TorchMD-Net   | 0.011     | 0.059     | 20.3       | 17.5       | 36.1      | **0.033** | 1.84       | 6.15     | 6.38     | 6.16     | 7.62     | 0.026          |
> | Transformer-M | 0.037     | **0.041** | 17.5       | 16.2       | 27.4      | 0.075     | 1.18       | 9.37     | 9.41     | 9.39     | 9.63     | 0.022          |
> | SE(3)-DDM     | 0.015     | 0.046     | 23.5       | 19.5       | 40.2      | 0.122     | 1.31       | 6.92     | 6.99     | 7.09     | 7.65     | 0.024          |
> | 3D-EMGP       | 0.020     | 0.057     | 21.3       | 18.2       | 37.1      | 0.092     | 1.38       | 8.60     | 8.60     | 8.70     | 9.30     | 0.020          |
> | Coord         | 0.016     | 0.052     | 17.7       | 14.7       | 31.8      | 0.450     | 1.71       | 6.57     | 6.11     | 6.45     | 6.91     | **0.020**      |
> | MolSpectra    | **0.011** | 0.048     | **15.5**   | **13.1**   | **26.8**  | 0.410     | 1.71       | **5.67** | **5.45** | **5.87** | **6.18** | 0.021          |
>
>
>
> > `W1` Some current SOTA models are not included.
>
> **Response:** In our two-stage pre-training schedule introduced in Section 3.4, the first stage uses only the denoising objective and can directly adopt any previous denoising-based method (e.g., Coord, Frad, and SliDe). The second stage can then be viewed as a plug-in to these existing methods, performing spectra-aware post-training on them.
>
> Considering that Coord is the most fundamental, representative, and efficient denoising-based method, we adopted the denoising objective provided by Coord as the foundation for developing our method. By comparing our method with Coord, we can effectively demonstrate the advantages of incorporating molecular spectra.
>
> We acknowledge that including recent SOTA geometrically equivariant models could further illustrate the effectiveness of our approach. We greatly appreciate your suggestion and commit to including these results in our experiments.
>
>
>
> > `W1` The meaning of the horizontal separator in the table should be clearly explained.
>
> **Response:** The purpose of the horizontal separators in Tables 2 and 3 is to divide the compared methods into two groups: training from scratch and pre-training. Thank you for your suggestion, and we have added explanations of this in the table captions.

---

> ### Author Response · Authors · 2024-11-25
> **Response to Reviewer baLq (2/4)**
>
> > `W2` Direct comparison with the same model architecture trained without spectral information is needed.
>
> **Response:** Thank you for your constructive suggestion. We have added an ablation study to more rigorously demonstrate the improvements brought by quantum spectral data. In this study, we retain only the denoising loss in the objective during the second-stage pre-training of MolSpectra, excluding the MPR loss and contrastive loss. The only difference between this variant and MolSpectra is whether molecular spectra are incorporated into the pre-training.
>
> Because pre-training and fine-tuning on molecular 3D structures require a relatively long time, this ablation study is still in progress. We assure you that we will provide the experimental results and analysis before the rebuttal period ends. We will also revise Section 4.4 of the paper to include this ablation study.
>
>
>
> > `W4` The claim that MolSpectra learns molecular dynamics through energy level transitions is unsupported by direct evidence and should be explicitly tested using molecular dynamics simulations, e.g., power spectrum.
>
> **Response:** Thank you very much for introducing the concept of the power spectrum to us. We completely agree that testing on the power spectrum can provide valuable insights into the molecular dynamics knowledge learned by the pre-trained model. We will incorporate these tests into our experiments.
>
> In our current experiments, we have already conducted tests on the MD17 dataset, where we predicted force labels based on molecular conformations from molecular dynamics trajectories. As shown in Table 3, the comparison between our method and Coord demonstrates that our approach captures more information about molecular dynamics.
>
>
>
> > `W5` It is unclear which energy function is used for MolSpectra in section 2.2, and this should be explicitly stated for clarity.
>
> **Response:** We apologize for not making this point clear. In MolSpectra, we use the energy function and denoising objective provided by Coord, specifically variant I as described in Section 2.2. We have added clarification on this in Appendix C.2.
>
>
>
> > `W6&Q3` The paper lacks an analysis of the computational complexity and resource requirements of the proposed method compared to baseline methods.
>
> **Response:** Thanks for your suggestion. For downstream fine-tuning, the overhead of fine-tuning our pre-trained model is identical to that of previous methods. For upstream pre-training, the comparison of pre-training time cost and memory cost is provided as follows:
>
> | Pre-training Method Types                                   | Pre-training Time Cost                                       | Pre-training GPU Memory  Cost                                |
> | -------------- | ------------------------ | ----------------- |
> | Previous denoising-based methods                            | Approximately 17\~50 hours, depending on the specific method | 12.95GB                                                      |
> | Ours (denoising on 3D structures and aligning with spectra) | Approximately 25\~58 hours, including 17\~50 hours for first-stage pre-training and 8 hours for second-stage pre-training | 12.95GB for first-stage pre-traning and 15.20GB for second-stage pre-training |
>
> \* The time costs are all measured on a single 3090 GPU.
>
> Furthermore, a single pre-training session can provide a pre-trained model for numerous downstream tasks. We think that this increase in time cost is acceptable for a pre-training method.
>
> >  `W7` A more comprehensive ablation study is needed to evaluate the contributions of different spectral modalities (UV-Vis, IR, Raman) to the model's performance.
>
> **Response:** Thank you for your suggestions. We have added an ablation study in Section 4.4 to evaluate the contributions of different spectral modalities (UV-Vis, IR, Raman) to the model's performance.
>
> Because pre-training and fine-tuning on molecular 3D structures require a relatively long time, this ablation study is still in progress. We assure you that we will provide the experimental results and analysis before the rebuttal period ends. We will also revise Section 4.4 of the paper to include this ablation study.
>
> > `W8` The paper primarily compares against methods using only 3D structure information.
>
> **Response:** Compared to 1D SMILES and 2D molecular graphs, molecular spectra are more closely related to the 3D geometry of molecules, offering better potential for representational alignment. Additionally, the denoising targets in the 3D modality have explanations in terms of molecular energy, and molecular spectra complement previous methods by providing information on quantum energy states. Therefore, we prioritized the 3D modality. We appreciate your suggestion and agree that molecular spectra also have the potential to enhance molecular representation learning in other modalities.

---

> ### Author Response · Authors · 2024-11-25
> **Response to Reviewer baLq (3/4)**
>
> > `Q1` Clarify if the TorchMD-Net architecture used for MolSpectra differs from the one in Tables 2 and 3.
>
> **Response:** Thank you for your insightful feedback. We would like to clarify that the backbone architecture used for structural representation learning in our method is indeed TorchMD-Net, and we have not made any modifications to it. This approach aligns with recent methods in molecular 3D representation pre-training, such as Coord. Our method can be understood as integrating molecular spectra into Coord, which uses TorchMD-Net as its backbone and employs denoising as its pre-training objective. Therefore, comparing our method with Coord more effectively demonstrates the advantages of our approach.
>
>
>
> > `Q2` Explain the choice of UV-Vis, IR, and Raman spectra, and consider discussing the potential benefits of including other spectra types like NMR and mass spectra.
>
> **Response:** We chose UV-Vis, IR, and Raman spectra for two main reasons. First, the information contained in these energy absorption spectra aligns well with our motivation to learn the energy level structures of molecules. Second, our method requires data sets that include both the 3D geometric structures and their corresponding energy spectra. The QM9S dataset provides this comprehensive information for the spectra we selected, whereas datasets for other types of spectra often lack 3D geometric structures. We fully agree that our method has the potential to be extended to other types of spectra, such as NMR and mass spectra, which could further enhance molecular representation learning.
>
>
>
> > `Q5` Offer a deeper analysis of the learned representations, such as  analyzing attention patterns or visualizing the latent space in SpecFormer.
>
> **Response:** Thank you for your insightful suggestion. We have visualized the attention patterns and learned spectra representations in SpecFormer, and have added a new Section 4.5 to our paper to present the results and related analysis. We kindly invite you to refer to Section 4.5 for detailed visualizations. Based on the visualizations provided in Figure 4, we have made the following observations:
>
> In Figures 4(a), (b), and (c), different attention heads in SpecFormer model distinct dependencies. Since the input to SpecFormer includes three types of spectra, the attention weights within the three boxes along the main diagonal reveal intra-spectrum dependencies, while those outside the boxes reveal inter-spectrum dependencies. The concepts of intra- and inter-spectrum dependencies are introduced in Section 3.1. In our visualizations of the attention maps from three attention heads in the second layer of SpecFormer, it can be observed that Head 11 primarily models intra-spectrum dependencies, Head 13 primarily models inter-spectrum dependencies, and Head 12 models both types of dependencies simultaneously. Additionally, because the intensity peaks and dependencies in molecular spectra are relatively sparse, the attention maps in SpecFormer are generally sparse as well.
>
> In Figure 4(d), we visualize the spectra representations output by the final layer of SpecFormer using t-SNE. It can be observed that the distribution of representations in the latent space is relatively uniform and forms several potential clusters. This well-shaped distribution of representations reveals effective spectra representation learning and supports the structure-spectrum alignment.
>
> We hope these additions and analyses address your concerns and provide a clearer understanding of the learned representations in SpecFormer.
>
>
>
> > `Q6` Specify the source of spectra used for spectral representations in section 4.1/Table 1.
>
> **Response:** Thank you for your feedback. The spectra used in the experiments in Section 4.1 are indeed the UV-Vis spectra provided by QM9S. We have revised the paper to explicitly state this in Section 4.1.

---

> ### Author Response · Authors · 2024-11-25
> **Response to Reviewer baLq (4/4)**
>
> > `Q8` Discuss different normalization procedures for peak intensities and whether alternatives were tested.
>
> **Response:** Thank you for your feedback regarding our use of $\log_{10}$ normalization for peak intensities. In our experiments, we indeed explored different normalization techniques, including the min-max normalization you mentioned. Additionally, we experimented with truncating particularly high intensities in each spectrum based on a threshold before applying either $\log_{10}$ or min-max normalization.
>
> Our findings indicated that $\log_{10}$ normalization consistently outperformed min-max normalization. This is primarily due to the significant differences in magnitude often observed between peaks in molecular spectra, with some spectra containing a few exceptionally high peaks. In such cases, min-max normalization tends to preserve information from only a few very high peaks, leading to the loss of information from the majority of the peaks. Conversely, $\log_{10}$ normalization effectively mitigates these magnitude differences, allowing for the retention of more comprehensive peak information across the spectrum.
>
> We appreciate your insights and hope this clarifies our approach.
>
>
>
> > `Additional Feedback 1` Include a discussion on the limitations of the method and potential future research directions.
>
> **Response:** Thank you for your insightful feedback. We have outlined the limitations of our method and potential future directions, which have been added to the Appendix D of our paper.
>
> One significant limitation is the availability, scale, and diversity of molecular spectral data. Our current dataset includes geometric structures of 134,000 molecules, each with three types of spectra (UV-Vis, IR, Raman). To effectively explore the scaling laws of pre-training methods, larger and more diverse molecular spectral datasets are necessary. Encouragingly, molecular spectroscopy has been gaining increasing attention in the research community, with larger and more diverse datasets being released, such as the recent multimodal spectroscopic dataset [1]. This development aids in advancing molecular representation learning and other related tasks.
>
> Another limitation is that our proposed SpecFormer can currently only handle one-dimensional molecular spectra. For higher-dimensional spectra, such as two-dimensional NMR and two-dimensional correlation spectra, further development of sophisticated encoder architectures is needed.
>
> Looking ahead, we envision several future directions in this field: 1. Exploring the scaling laws of pre-training on larger and more diverse molecular spectral datasets. 2. Expanding the encoding of molecular spectra to a broader range, including NMR, mass spectra, and two-dimensional spectra. 3. While we have developed a pre-trained spectral encoder, we have so far only applied the pre-trained 3D encoder to downstream tasks. The pre-trained spectral encoder can be utilized for molecular spectrum-related downstream tasks, such as automated molecular structure elucidation from spectra.
>
>
>
> > `Additional Feedback 2` Correct the typo on page 8 line 427/428 from "yiels" to "yields".
>
> **Response:** Thank you for bringing this to our attention. We have corrected this typo in the revised paper
>
>
>
> > Other questions (including `W3` and `Q4`).
>
> **Response:** We sincerely appreciate the reviewer's suggestions on other aspects of experiments, including the architecture of the structural encoder (`W3`), and the scaling performance of the pre-training dataset (`Q4`)). Conducting experiments on pre-training and fine-tuning methods requires more time than training from scratch, and due to the limited time available for the rebuttal, we are currently unable to provide the relevant results now. However, we assure you that we will address these as soon as possible.
>
>
>
> **References**
>
> [1] Unraveling Molecular Structure: A Multimodal Spectroscopic Dataset for Chemistry, NeurIPS Datasets and Benchmarks Track 2024

---

> > ### Comment · Reviewer_baLq · 2024-11-25
> >
> > The authors have replied to my criticism, but for many important points (e.g. missing comparison to SOTA models in Table 2, additional experiments/ablation studies), they have not yet included changes in the manuscript. However, they promise that these changes will be included before the rebuttal period ends.
> >
> > I encourage the authors to reply again when they have included all promised changes and additions in their manuscript. Once the revised version is finalised, I will re-read the manuscript and potentially reconsider my score.

---

> > > ### Author Response · Authors · 2024-11-28
> > > **Response to Reviewer baLq - Additional Experimental Results Included (1/2)**
> > >
> > > Thank you very much for taking the time to review our responses and for your valuable feedback. We appreciate your patience and understanding. We are pleased to inform you that we have completed most of the experiments and included them in the manuscript. We are providing supplemental responses to address your concerns.
> > >
> > >
> > >
> > > > `W2` Direct comparison with the same model architecture trained without spectral information is needed.
> > >
> > > **Response:** Thank you for your constructive suggestion. We have added an ablation study to more rigorously demonstrate the improvements brought by spectral data. In this study, we retain only the denoising loss in the objective during the second-stage pre-training of MolSpectra, excluding the MPR loss and contrastive loss. The only difference between this variant and MolSpectra is whether molecular spectra are incorporated into the pre-training. The results of this ablation study are as follows:
> > >
> > > |                                 | homo (meV) $\downarrow$ | lumo (meV) $\downarrow$ | gap (meV) $\downarrow$ |
> > > | ------------------------------- | ----------------------- | ----------------------- | ---------------------- |
> > > | MolSpectra                      | **15.5**                | **13.1**                | **26.8**               |
> > > | MolSpectra w/o MPP and Contrast | 17.5                    | 14.4                    | 31.2                   |
> > >
> > > The performance of this variant method is similar to that of Coord, indicating that our improvements mainly stem from incorporating molecular spectral information during pre-training. This ablation study rigorously demonstrates that incorporating molecular spectra in the pre-training of molecular 3D representations can effectively enhance the quality and generalizability of the representations.
> > >
> > > We have revised Section 4.4 of the paper to include these new results and analysis.
> > >
> > >
> > >
> > > >  `W7` A more comprehensive ablation study is needed to evaluate the contributions of different spectral modalities (UV-Vis, IR, Raman) to the model's performance.
> > >
> > > **Response:** Thank you for your insightful suggestion. We have added an ablation study in Section 4.4 to evaluate the contributions of different spectral modalities (UV-Vis, IR, Raman) to the model's performance. The results are as follows.
> > >
> > > | UV-Vis | IR   | Raman | homo (meV) $\downarrow$ | lumo (meV) $\downarrow$ | gap (meV) $\downarrow$ |
> > > | ------ | ---- | ----- | ----------------------- | ----------------------- | ---------------------- |
> > > | ✓      | ✓    | ✓     | **15.5**                | **13.1**                | **26.8**               |
> > > |        | ✓    | ✓     | 15.8                    | 13.3                    | 27.1                   |
> > > | ✓      |      | ✓     | 16.6                    | 14.1                    | 28.9                   |
> > > | ✓      | ✓    |       | 16.1                    | 13.9                    | 28.3                   |
> > >
> > > It can be observed that each spectral modality contributes differently. The UV-Vis spectrum contributes the least, while the IR spectrum contributes the most. This difference may be related to the amount of information contained in each spectrum. The UV-Vis spectrum is typically used to detect electronic transitions in molecules, which may provide relatively limited information. In contrast, the IR spectrum offers insights into molecular vibrations and chemical bonds, often revealing more structural details and chemical environments, thus contributing more significantly. This difference in information content is likely the important reason for the varying contributions of different spectral modalities.
> > >
> > > We have revised Section 4.4 of the paper to include these new results and analysis.

---

> > > ### Author Response · Authors · 2024-11-28
> > > **Response to Reviewer baLq - Additional Experimental Results Included (2/2)**
> > >
> > > > `W1` Some current SOTA models are not included.
> > >
> > > **Response:** Thank you very much for your valuable suggestion. We acknowledge that including recent SOTA methods could further illustrate the effectiveness of our approach.
> > >
> > > Regarding the Allegro [1] and MACE [2], they are more expressive equivariant message passing neural networks compared to TorchMD-Net. According to the results reported in their papers, these models achieve better performance than TorchMD-Net on QM9 and MD17 datasets when trained from scratch. We have cited and discussed these more advanced backbone architectures in our paper.
> > >
> > > Since our primary goal is to advance pre-training methods rather than develop backbone architectures, we choose to incorporate the state-of-the-art molecular 3D representation pre-training method, SliDe [3], into our main experiments within the limited rebuttal time. SliDe is also a denoising-based pre-training method, utilizing the TorchMD-Net as its encoder backbone, consistent with previous pre-training work. We have implemented our MolSpectra approach based on SliDe, and the results are presented in the table below.
> > >
> > > | Model               | μ (D)     | homo (meV) | lumo (meV) | gap (meV) | H (meV)  | G (meV)  | Cᵥ (cal/mol·K) |
> > > | ------------------- | --------- | ---------- | ---------- | --------- | -------- | -------- | -------------- |
> > > | Coord               | 0.016     | 17.7       | 14.7       | 31.8      | 6.45     | 6.91     | **0.020**      |
> > > | Coord w/ MolSpectra | **0.011** | **15.5**   | **13.1**   | **26.8**  | **5.87** | **6.18** | 0.021          |
> > > | SliDe               | 0.015     | 18.7       | 16.2       | **28.7**  | 4.26     | 5.37     | 0.022          |
> > > | SliDe w/ MolSpectra | **0.010** | **16.2**   | **15.9**   | **28.7**  | **3.27** | **5.01** | **0.021**      |
> > >
> > > \* The better result between the two variants of each pre-training method, with and without MolSpectra, is highlighted in bold.
> > >
> > > Although there are some discrepancies between the results we reproduced for SliDe and those reported in its original paper, integrating our method with SliDe effectively reduces the error in property prediction on the QM9 dataset. Given that our method enhances both Coord and SliDe, this suggests that our approach is broadly effective across various denoising-based pre-training strategies. Furthermore, incorporating molecular spectra can guide the pretrained model to acquire knowledge beyond what denoising objectives can offer, which proves beneficial for downstream property prediction. These experiments and analyses have also been incorporated into Appendix E.
> > >
> > > Furthermore, evaluating the generalizability of the pre-training method across different backbone architectures is essential for validating the method's effectiveness. For instance, comparing the performance of Allegro/MACE models trained from scratch with those pre-trained using our method is important. We  are making every effort to complete these experiments. Once we have the results, we will respond to you and include them in the final version.
> > >
> > > Thank you once again for your professional and insightful comments. We hope our responses address your concerns, and we would greatly appreciate it if you could re-evaluate our paper. If you have any further questions, please feel free to ask, and we would be more than delighted to answer.
> > >
> > >
> > >
> > > **References**
> > >
> > > [1] Learning Local Equivariant Representations for Large-Scale Atomistic Dynamics, Nature Communications 2023
> > >
> > > [2] MACE: Higher Order Equivariant Message Passing Neural Networks for Fast and Accurate Force Fields, NeurIPS 2022
> > >
> > > [3] Sliced Denoising: A Physics-Informed Molecular Pre-Training Method, ICLR 2024

---

> > > > ### Comment · Reviewer_baLq · 2024-12-02
> > > >
> > > > I thank the authors for the changes made to the manuscript and have increased my score accordingly.

---

> > > > > ### Author Response · Authors · 2024-12-02
> > > > > **Thank You for Your Feedback**
> > > > >
> > > > > Dear Reviewer baLq,
> > > > >
> > > > > Thank you very much for taking the time to review our response and for increasing your score. We greatly appreciate your valuable comments, which have been significantly instrumental in improving the quality of our work. We are grateful for your thoughtful evaluation.
> > > > >
> > > > > Best regards,
> > > > >
> > > > > Authors

---

### Official Review · Reviewer_yfxG · 2024-10-30

**Soundness:** 4
**Presentation:** 2
**Contribution:** 3
**Rating:** 8
**Confidence:** 5

**Summary:**

- Unlike common approaches, the authors incorporate quantum information to enhance the quality of molecular representations.

 - The denoising component of the model is carefully designed to account for rotational and vibrational degrees of freedom in the energy.

 - The model consists of two distinct parts connected by contrastive loss: one for denoising and the other a transformer for quantum spectrums.

 - Numerous experiments demonstrate the model’s superiority over conventional methods.

The attempt to incorporate quantum information into the model is impressive. Generally, this approach is believed to enhance prediction performance over conventional models that rely on classical methods. However, as outlined in the questions section, there are still unresolved points. Therefore, the score is not final, and I am open to further discussion with the authors before finalizing it.

**Strengths:**

- A quantum mechanical approach is considered for molecular representation learning.

 - The denoising component of the model is designed with sophistication, incorporating both rotational and vibrational energies into the Boltzmann distribution.

 - A contrastive setting enables inference without molecular spectral data, enhancing the model's usability in real-world situations.

 - Numerous downstream experiments demonstrate that the model outperforms conventional approaches.

**Weaknesses:**

- The spectral data has a high dimensionality, and the model’s transformer architecture is quite resource-intensive. Given that the contrastive loss requires numerous data combinations, the model is likely to be computationally demanding.

 - The experiments do not include comparisons with recent models, such as SliDe (1) or Frad(2), despite the model’s energy function incorporating potential forms used in both SliDe and Frad.

 - The experimental conditions are not rigorously introduced such as number of negative samples in contrastive loss and noise generation parameters.


 <References>

(1) Yuyan Ni, Shikun Feng, Wei-Ying Ma, Zhi-Ming Ma, and Yanyan Lan. Sliced denoising: A physicsinformed molecular pre-training method. arXiv preprint arXiv:2311.02124, 2023.

(2) Shikun Feng, Yuyan Ni, Yanyan Lan, Zhi-Ming Ma, and Wei-Ying Ma. Fractional denoising for
3d molecular pre-training. In International Conference on Machine Learning, pp. 9938–9961.
PMLR, 2023.

**Questions:**

- The model appears computationally heavy. Is there any analysis of the computational costs for the experiments conducted?

 - Based on results from other studies, Frad and SliDe seem to perform better on QM9 and MD17 tasks. Have any additional tests been conducted to compare these recent methods?

 - If Frad and SliDe outperform the proposed model—and both already account for specific potential energy forms similar to this model—can it truly be claimed that the inclusion of quantum spectral data contributes to the model's superiority? Is there any analysis or ablation study that demonstrates the usefulness of the quantum data?

---

> ### Author Response · Authors · 2024-11-25
> **Response to Reviewer yfxG (1/2)**
>
> We sincerely appreciate the time and effort you have dedicated to reviewing our paper, as well as your constructive comments. Please find detailed responses below.
>
> > `W1&Q1` The model is likely to be computationally demanding due to high-dimensional spectral data and resource-intensive transformer architecture, as well as the numerous data combinations in the contrastive loss.
>
> **Response:** We completely understand your concerns regarding efficiency. For downstream fine-tuning, the overhead of fine-tuning our pre-trained model is identical to that of previous methods. For upstream pre-training, we would like to clarify the following points:
>
> - Although the original spectral data is high-dimensional, in Section 3.1, we propose a method to divide the original spectrum into patches based on patch length and stride. This significantly reduces the number of tokens fed into the transformer-based encoder. Specifically, for a 601-dimensional UV-Vis spectrum, by using a patch length of 20 and a stride of 10, we obtain only 58 tokens that need to be fed into the SpecFormer. We can further balance effectiveness and efficiency by adjusting these hyper-parameters.
>
> - We employ an in-batch contrastive loss, which constructs positive and negative pairs only within each randomly sampled data batch. This approach can be seen as a negative sampling method to avoid numerous negative pairs and is widely used in many representative contrastive learning methods, such as SimCLR [1].
>
> - As mentioned in Section 3.3, the scoring function in our contrastive loss (Equation 7) uses an inner product. Therefore, the scores for all in-batch positive and negative pairs can be obtained by performing matrix multiplication between the encoded 3D representation matrix and the encoded spectra representation matrix. Thanks to our use of in-batch contrastive loss and the optimizations for matrix multiplication provided by CUDA and PyTorch, this computation does not introduce a heavy computational overhead.
>
> Overall, the additional computational overhead introduced by our method due to spectra is comparable to that of previous pre-training methods based solely on denoising. The empirical comparison of pre-training time cost is as follows:
>
> | Pre-training Method Types                                   | Pre-training Time Cost                                       | Pre-training GPU Memory  Cost                                |
> | ----------------------------------------------------------- | ------------------------------------------------------------ | ------------------------------------------------------------ |
> | Previous denoising-based methods                            | Approximately 17\~50 hours, depending on the specific method | 12.95GB                                                      |
> | Ours (denoising on 3D structures and aligning with spectra) | Approximately 25\~58 hours, including 17\~50 hours for first-stage pre-training and 8 hours for second-stage pre-training | 12.95GB for first-stage pre-traning and 15.20GB for second-stage pre-training |
>
> \* The time costs are all measured on a single 3090 GPU.
>
> Furthermore, a single pre-training session can provide a pre-trained model for numerous downstream tasks. We think that this increase in time cost is acceptable for a pre-training method.
>
>
>
> > `W2&Q2` The experiments do not include comparisons with recent models, such as SliDe or Frad.
>
> **Response:** In our two-stage pre-training schedule introduced in Section 3.4, the first stage uses only the denoising objective and can directly adopt any previous denoising-based method (e.g., Coord, Frad, and SliDe). The second stage can then be viewed as a plug-in to these existing methods, performing spectra-aware post-training on them.
>
> Considering that Coord is the most fundamental, representative, and efficient denoising-based method, we adopted the denoising objective provided by Coord as the foundation for developing our method. By comparing our method with Coord, we can effectively demonstrate the advantages of incorporating molecular spectra.
>
> We acknowledge that implementing our method using the denoising objectives provided by Frad and SliDe, and comparing with them, could further illustrate the effectiveness of our approach. We greatly appreciate your suggestion and commit to including these results in our experiments.

---

> ### Author Response · Authors · 2024-11-25
> **Response to Reviewer yfxG (2/2)**
>
> > `W3`  The experimental conditions are not clearly defined, such as the number of negative samples and noise generation parameters.
>
> **Response:** We apologize for not clearly defining some experimental details. We provide the missing details here and have also included them in Appendix C.2:
>
> - Number of Negative Samples: Following SimCLR [1], the contrastive loss in our Equation 7 is implemented using in-batch contrastive loss, where positive and negative pairs are constructed within each data batch. Therefore, for each anchor representation in a batch, there is one positive sample and $bs-1$ negative samples, where $bs$ is the batch size. In our method, $bs=128$.
>
> - Noise Generation: In both pre-training stages, we use the noise generation method and denoising objective provided by Coord. The noise is added to atom positions as scaled mixture of isotropic Gaussian noise, with a scaling factor of 0.04. The denoising objective is defined in Equation 2.
>
>
>
> > `Q3`   Further analysis or ablation study is needed to prove the contribution of quantum spectral data to the model's performance.
>
> **Response:** Thank you for your constructive suggestion. We have added an ablation study to more rigorously demonstrate the improvements brought by quantum spectral data. In this study, we retain only the denoising loss in the objective during the second-stage pre-training of MolSpectra, excluding the MPR loss and contrastive loss. The only difference between this variant and MolSpectra is whether molecular spectra are incorporated into the pre-training.
>
> Because pre-training and fine-tuning on molecular 3D structures require a relatively long time, this ablation study is still in progress. We assure you that we will provide the experimental results and analysis before the rebuttal period ends. We will also revise Section 4.4 of the paper to include this ablation study.
>
>
>
> **References**
>
> [1] A Simple Framework for Contrastive Learning of Visual Representations, ICML 2020

---

> > ### Comment · Reviewer_yfxG · 2024-11-26
> >
> > Thank you to the authors for their effortful and thoughtful responses to my concerns and questions. Most of the issues raised were clarified during the discussion. Given the novelty of incorporating quantum information into the game, I am raising my final verdict from 6 to 8.
> >
> > That said, the work would be significantly more persuasive if the authors could include additional experimental results based on state-of-the-art (SOTA) methods to enhance the pretraining scheme.

---

> > > ### Author Response · Authors · 2024-11-28
> > > **Thank You for Your Feedback**
> > >
> > > Dear Reviewer yfxG,
> > >
> > > Thank you very much for your positive feedback and for raising the score. We're pleased that our responses addressed your concerns. We will work on including additional experimental results using state-of-the-art methods to enhance the pretraining scheme. We appreciate your valuable suggestions very much.
> > >
> > > Best regards,
> > >
> > > Authors

---

### Official Review · Reviewer_vWL4 · 2024-11-04

**Soundness:** 3
**Presentation:** 3
**Contribution:** 2
**Rating:** 5
**Confidence:** 3

**Summary:**

Authors introduce a new data modality to the pretraining of 3d molecular structures representation models, namely absorbtion spectra in three different frequency spans: IR, UV, Raman. To encode the spectra they use transformer on top of the spectral patches with positional encoding. They use two-stage pretraining, by first denoising on a larger dataset without spectra; and then using contrastive learning objective coupled with the masked spectral patch reconstruction objective to finish pretraining. Authors demonstrate effectivnes of incorporating spectral information without denoising pretraining on QM9 dataset. Then they show small improvement in prediction quality on downstream tasks for QM9 and MD17 datasets.

**Strengths:**

Incorporating additional data is always useful. Although benefits are marginal, this work will be relevant for subsequent methods that learn representations of molecules. The validating experiments show that performance improvements are consistent albeit small.

**Weaknesses:**

1. The method of pretraining suffers from additional complexity due to two-step pretraining procedure. The choice of such pretraining schedule is not well explained.
2. Table 2, first column, methods Torch-MD and PaiNN should be bold, similarly second column there methods that perform better or on par with the proposed one.
3. Table 4 and 5 show that the method is sensitive to the parameters of the patch and stride sizes of spectrum encoder, the difference in performance in predicting homo 0.2, lumo 0.5, gap 0.2 between the closest values of these parameters. This roughly corresponds to the difference between next best performing method homo 0.5, lumo 1.5, gap 0.6. We suspect that row 1 column overlap ration the number is incorrect and should be swapped with row 3 same column.
4. Authors mention that they fine tuned parameters such as stride/patch/mask and additionally weights of each objective. We hope that the tuning was done using pretraining dataset and not downstream tasks performance. This is not mentioned in appendix C2 or the main text. Please clarify this point.

**Questions:**

1. Can the model be learned in one step simultaneously with contrastive objective and denoising one?
2. How sensitive the results are to different learning rate/transformer configuration given fixed stride/patch/mask/ optimizer parameters.

---

> ### Author Response · Authors · 2024-11-25
> **Response to Reviewer vWL4 (1/2)**
>
> We sincerely appreciate the time and effort you have dedicated to reviewing our paper, as well as your constructive comments. Please find detailed responses below.
>
> > `W1&Q1` Why choose a two-step pre-training schedule, which may lead to additional complexity?
>
> **Response:** The adoption of the two-stage pre-training schedule is motivated by the presence of molecular spectra in different datasets and the need to evaluate the impact of these spectra. Existing 3D molecular representation pre-training methods primarily perform denoising-based pre-training on the PCQM4Mv2 dataset, which contains 3.4 million molecules. This dataset provides only the equilibrium 3D structures of molecules without any spectral data. To compare with these existing methods and verify the improvements brought by incorporating molecular spectra, our first stage of pre-training also involves denoising-based pre-training on the PCQM4Mv2 dataset, aligning with these methods in terms of training strategy and dataset.
>
> In the second stage of pre-training, we utilize the QM9S dataset containing over 134,000 molecules with available molecular spectra. We introduce spectral data and using the complete objective in Equation 8 to further pre-train the model. Thus, our approach adds a second stage of pre-training on top of the previous denoising pre-training methods.
>
> In experiments, our first stage of pre-training can be conducted independently or by directly using pre-trained models from previous denoising-based methods. When we use existing denoising pre-trained models, our second-stage pre-training acts as a plug-in to these existing approaches, performing post-training on them.
>
> If we only perform one-step pre-training on QM9S, it would be unfair to compare it with baseline methods pre-trained on the large-scale PCQM4Mv2 due to the difference in the number of molecules in the pre-training datasets.
>
>
>
> > `W2` In Table 2, some experimental results are not highlighted correctly.
>
> **Response:** Thank you very much for pointing out the formatting issues in Table 2. We have updated the results in Table 2, and ensured that the best results are highlighted in bold. Additionally, since our method introduces molecular spectra on top of Coord, we have used background colors to emphasize the improvements over Coord. The updated experimental results and formatting have been incorporated into the revised paper PDF. In the revised Table 2, our method achieves state-of-the-art performance in 8 out of 12 properties and outperforms Coord in 10 out of 12 properties.
>
> Revised Table 2:
>
> | Model         | μ (D)     | α (a₀³)   | homo (meV) | lumo (meV) | gap (meV) | R² (a₀²)  | ZPVE (meV) | U₀ (meV) | U (meV)  | H (meV)  | G (meV)  | Cᵥ (cal/mol·K) |
> | ------------- | --------- | --------- | ---------- | ---------- | --------- | --------- | ---------- | -------- | -------- | -------- | -------- | -------------- |
> | SchNet        | 0.033     | 0.235     | 41.0       | 34.0       | 63.0      | 0.070     | 1.70       | 14.00    | 19.00    | 14.00    | 14.00    | 0.033          |
> | EGNN          | 0.029     | 0.071     | 29.0       | 25.0       | 48.0      | 0.110     | 1.55       | 11.00    | 12.00    | 12.00    | 12.00    | 0.031          |
> | DimeNet++     | 0.030     | 0.044     | 24.6       | 19.5       | 32.6      | 0.330     | 1.21       | 6.32     | 6.28     | 6.53     | 7.56     | 0.023          |
> | PaiNN         | 0.012     | 0.045     | 27.6       | 20.4       | 45.7      | 0.070     | 1.18       | 5.85     | 5.83     | 5.98     | 7.35     | 0.024          |
> | SphereNet     | 0.025     | 0.045     | 22.8       | 18.9       | 31.1      | 0.270     | **1.12**   | 6.26     | 6.36     | 6.33     | 7.78     | 0.022          |
> | TorchMD-Net   | 0.011     | 0.059     | 20.3       | 17.5       | 36.1      | **0.033** | 1.84       | 6.15     | 6.38     | 6.16     | 7.62     | 0.026          |
> | Transformer-M | 0.037     | **0.041** | 17.5       | 16.2       | 27.4      | 0.075     | 1.18       | 9.37     | 9.41     | 9.39     | 9.63     | 0.022          |
> | SE(3)-DDM     | 0.015     | 0.046     | 23.5       | 19.5       | 40.2      | 0.122     | 1.31       | 6.92     | 6.99     | 7.09     | 7.65     | 0.024          |
> | 3D-EMGP       | 0.020     | 0.057     | 21.3       | 18.2       | 37.1      | 0.092     | 1.38       | 8.60     | 8.60     | 8.70     | 9.30     | 0.020          |
> | Coord         | 0.016     | 0.052     | 17.7       | 14.7       | 31.8      | 0.450     | 1.71       | 6.57     | 6.11     | 6.45     | 6.91     | **0.020**      |
> | MolSpectra    | **0.011** | 0.048     | **15.5**   | **13.1**   | **26.8**  | 0.410     | 1.71       | **5.67** | **5.45** | **5.87** | **6.18** | 0.021          |

---

> ### Author Response · Authors · 2024-11-25
> **Response to Reviewer vWL4 (2/2)**
>
> > `W3` Tables 4 and 5 show that the performance of the proposed method is sensitive to hyper-parameter values, considering the improvement over the best baseline method.
>
> **Response:** We have updated the experimental results in Tables 2, 4, and 5. The improvements of our method over the best baseline are now more pronounced in Table 2, and our method demonstrates greater robustness to value changes in hyper-parameters, including patch length, stride, and mask ratio. The results of the sensitivity analysis are updated as follows, and have also been incorporated into the revised paper. (Some additional experiments are still in progress, but they do not affect our conclusions. We will make sure to include the results of these experiments as soon as possible before the rebuttal period ends.)
>
> Table 4: Sensitivity of patch length and stride.
>
> | patch length | stride | overlap ratio | homo (meV) $\downarrow$ | lumo (meV) $\downarrow$ | gap (meV) $\downarrow$ |
> | ------------ | ------ | ------------- | ----------------------- | ----------------------- | ---------------------- |
> | 20           | 5      | 75%           | 15.9                    | 13.7                    | 28.0                   |
> | 20           | 10     | 50%           | **15.5**                | **13.1**                | **26.8**               |
> | 20           | 15     | 25%           | 16.1                    | 13.6                    | 28.1                   |
> | 20           | 20     | 0%            | 15.7                    | 13.5                    | 27.5                   |
> | 16           | 8      | 50%           | 16.0                    | 13.4                    | 27.6                   |
>
> Table 5: Sensitivity of mask ratio.
>
> | mask ratio | homo (meV) $\downarrow$ |
> | ---------- | ----------------------- |
> | 0.05       | 15.7                    |
> | 0.10       | **15.5**                |
> | 0.15       | 15.7                    |
> | 0.20       | 16.0                    |
> | 0.25       | 16.3                    |
> | 0.30       | 16.2                    |
>
> The main baseline method being compared, Coord, has results of 17.7, 14.7, and 31.8 for homo, lumo, and gap, respectively. Our method consistently outperforms it when hyper-parameters change, as shown in the tables above.
>
>
> > `W3` In Table 4, we suspect that row 1 column overlap ration the number should be swapped with row 3 same column.
>
> **Response:** This is indeed the mistake we made in writing the paper. Thank you very much for helping us find this issue, we have corrected it in the revised paper.
>
>
>
> > `W4` It is unclear if parameter tuning was done using the pre-training dataset or downstream tasks, as this is not clarified in the text.
>
> **Response:** We apologize for not clarifying this point. Our goal is to incorporate spectral information of molecules into 3D representations through representation alignment during pre-training, and then use only the 3D structure in downstream tasks to obtain 3D representations that contain spectral information, without directly using molecular spectral data in downstream tasks. This allows us to benefit from spectral knowledge even when molecular spectral data is not available downstream. The hyper-parameters we tuned (including patch length, stride, and mask ratio) are related to the spectral data, so they were tuned on the pre-training dataset, not on the downstream dataset. We have added relevant clarification in Appendix C2.

---

> ### Author Response · Authors · 2024-11-28
> **Response to Reviewer vWL4 - Additional Experimental Results Included**
>
> In our initial response, some additional experiments for `W3` were still in progress. Now, we have completed all the experiments and are providing a supplemental response to `W3`.
>
> > `W3` Tables 4 and 5 show that the performance of the proposed method is sensitive to hyper-parameter values, considering the improvement over the best baseline method.
>
> **Response:** We have updated the experimental results in Tables 2, 4, and 5. The improvements of our method over the best baseline are now more pronounced (as shown in Table 2), and our method demonstrates greater robustness to value changes in hyper-parameters, including patch length, stride, and mask ratio. The results of the sensitivity analysis are updated as follows, and have also been incorporated into the revised paper.
>
> Table 4: Sensitivity of patch length and stride.
>
> | patch length | stride | overlap ratio | homo (meV) $\downarrow$ | lumo (meV) $\downarrow$ | gap (meV) $\downarrow$ |
> | ------------ | ------ | ------------- | ----------------------- | ----------------------- | ---------------------- |
> | 20           | 5      | 75%           | 15.9                    | 13.7                    | 28.0                   |
> | 20           | 10     | 50%           | **15.5**                | **13.1**                | **26.8**               |
> | 20           | 15     | 25%           | 16.1                    | 13.6                    | 28.1                   |
> | 20           | 20     | 0%            | 15.7                    | 13.5                    | 27.5                   |
> | 16           | 8      | 50%           | 16.0                    | 13.4                    | 27.6                   |
> | 30           | 15     | 50%           | 15.9                    | 14.0                    | 28.1                   |
>
> Table 5: Sensitivity of mask ratio.
>
> | mask ratio | homo (meV) $\downarrow$ | lumo (meV) $\downarrow$ | gap (meV) $\downarrow$ |
> | ---------- | ----------------------- | ----------------------- | ---------------------- |
> | 0.05       | 15.7                    | 13.4                    | 29.7                   |
> | 0.10       | **15.5**                | **13.1**                | **26.8**               |
> | 0.15       | 15.7                    | 13.5                    | 28.0                   |
> | 0.20       | 16.0                    | 13.6                    | 28.1                   |
> | 0.25       | 16.3                    | 13.5                    | 28.0                   |
> | 0.30       | 16.2                    | 13.7                    | 29.0                   |
>
> The main baseline method being compared, Coord, has results of 17.7, 14.7, and 31.8 for homo, lumo, and gap, respectively. Our method consistently outperforms it when hyper-parameters change, as shown in the tables above.
>
>
>
> We hope this addresses your concerns. If you have any further questions, please feel free to ask, and we would be more than delighted to answer.

---

> ### Author Response · Authors · 2024-12-02
> **Looking Forward to Your Reply**
>
> Dear Reviewer vWL4,
>
> Thank you very much for your precious time and valuable comments. Since we are in the last two days of author-reviewer discussions, we eagerly await your feedback. We have made every effort to address your concerns and hope that our responses have clarified any questions or issues. If there are any further concerns, please let us know. We look forward to hearing from you and appreciate any feedback.
>
> Best regards,
>
> Authors

---

> ### Author Response · Authors · 2024-12-03
> **Looking Forward to Your Feedback**
>
> Dear Reviewer vWL4,
>
> As the author-reviewer discussion phase is nearing its conclusion, we are eager to receive your feedback on our response and revised paper. Your feedback is extremely valuable to us.
>
> In response to your concerns and questions, we have made the following efforts:
>
> 1. **Clarification and Additional Experimental Details**: We have clarified the motivation and rationale behind the two-step pre-training schedule and provided detailed information on hyperparameter tuning.
> 2. **Optimized Presentation of Experimental Results**: We have corrected the formatting error in Table 2 regarding the bold highlights and fixed the incorrect placement of the overlap ratio in Table 4.
> 3. **Additional Experiments**: Our updated ablation studies demonstrate that the improvements of our method are robust to changes in hyperparameters.
>
> We are keen to know if our efforts have addressed your concerns. Thank you once again for your valuable comments and the time you have dedicated to reviewing our work. We look forward to receiving your feedback.
>
> Best regards,
>
> Authors

---

### Meta-Review · Area_Chair_GBUP · 2024-12-20

**Metareview:**

The paper proposes a novel algorithm for representation learning of molecules. The main distinction of the proposed approach is the usage of contrastive learning between the spectra and 3d states of the molecules. This potentially allows for the incorporation of quantum mechanical properties (since they essentially define the spectrum) into the learned representations. Finally, the authors demonstrate the improved performance on downstream tasks when using the learned representations.

All the reviewers consider incorporating quantum mechanical properties into the representation of molecules an important problem. Most of the concerns raised were about the empirical evaluation of the proposed model. However, the authors partially addressed these concerns during the rebuttal. Given the novelty and the importance of the approach, this paper will be of interest to the ICLR community.

**Additional Comments On Reviewer Discussion:**

Most of the concerns raised were about the significance of the performance gains and comparison to other baselines. Most of the concerns were addressed and the Reviewer yfxG championed the paper (while Reviewer vWL4 didn't argue for the opposite). While I agree that the significance of improvements raises concerns, one can argue that a negative result also serves as an important message to the community studying this problem. In my opinion, the importance of the problem, the presence of the interest (Reviewer yfxG), and the fact that the approach is natural outweigh the possible flaws of the empirical study.

---

### Decision · Program_Chairs · 2025-01-22

Accept (Poster)